# Investigation of the Robust Integration of Distributed Fibre Optic Sensors in Structural Concrete Components

**DOI:** 10.3390/s24186122

**Published:** 2024-09-22

**Authors:** Johannes Wimmer, Thomas Braml

**Affiliations:** Institute of Structural Engineering, University of the Bundeswehr Munich, 85577 Neubiberg, Germany; thomas.braml@unibw.de

**Keywords:** distributed fibre optical sensor, reinforcing steel, reinforced concrete, post-tensioning, regression analysis, DFOS, DOFS, distributed sensing, structural health monitoring

## Abstract

In recent times, the value of data has grown. This tendency is also observeable in the construction industry, where research and digitalisation are increasingly oriented towards the collection, processing and analysis of different types of data. In addition to planning data, measurement data is a main focus. fibre optic measurements offer a highly precise and comprehensive approach to data collection. It is, however, important to note that this technology is still in research regarding concrete structures. This paper presents two methods of integrating filigree sensors into concrete structures. The first approach entails wrapping a fibre around a tendon duct and analysing the installation and associated measurements. The second method involves bonding polyimide and acrylate-coated fibres with 2K epoxy and cyanoacrylate in the grooves of rebars, exposing them to chemical environments. The resulting measurement data is evaluated qualitatively and quantitatively to ascertain its resilience to environmental factors. These developed criteria are consolidated in a decision matrix. Fibre-adhesive combinations necessitate protection from chemical and mechanical influences. The limitations of the solutions are pointed out, and alternative options are proposed.

## 1. Introduction

Detecting strain in concrete structures is a challenge due to the inelastic material behaviour and the harsh environment during construction. Various sensors, such as strain gauges, linear displacement transducers or even potentiometers have been used for decades. They measure precisely but discretely. Hence, they are not well suited for finding cracks or strain peaks in concrete structures. New research in the field of fibre optics delivered new measurement methods. First approaches based on Fibre Bragg gratings (FBG) still only provided discrete measurement values. There, gratings are imprinted into an optical fibre. The aligning of gratings to form a measurement chain (e.g., in Kenel et al. [1]) resulted in the first high-resolution quasi-continuous strain measurements on fibres. As the imprinting of the grids is expensive, the further development of measurement methods of different back-scattering was target-oriented. Thus, the use of distributed fibre optic strain measurement (DFOS) found application [2]. Bado and Casas [3] showed an overview of the current developments in DFOS for civil engineering structural health monitoring. They cited Barrias et al. [4], who presented the technologies Raman optical time domain reflectometry (OTDR), Brillouin OTDR (BOTDR), Brillouin optical time domain analysis (BOTDA), Rayleigh optical frequency domain reflectometry (OFDR) and FBG. The summary in Table 1 is extracted from there. All presented technologies can be used for temperature measurement and all except Raman OTDR can be used additionally or alternatively for strain measurement. FBG also allows displacement measurement and Rayleigh distributed acoustic sensing (DAS) can be used to measure vibration.

Table 1 shows that the Rayleigh OFDR method is particularly suitable for measuring strain on small structures due to its fine resolution. This is done using Rayleigh backscattering of light, which is caused by refractive index and density fluctuations in the fibre. Basic information on this can be found in textbooks, such as [5]. The backscattering is recorded in a detector and the signal is evaluated. With these comparatively new methods, many investigations have already been carried out on the selection of the fibre or cable used, the bonding and the host material. Tests on the mechanical and chemical resistance of the sensors regarding durability show promising results.

In the second chapter we describe the current state of research on the design of DFOS for measuring strain on reinforced concrete structures. The Section 3 presents the key considerations for embedding fibres in concrete structures. The Section 4 describes the first approach on embedding with ducts for tendons and its results, Section 5 the experimental setup and results of experiments on rebars where the fibres were bonded into grooves. The conclusions from Section 3, Section 4 and Section 5 are discussed in Section 6, and Section 7 summarises the paper and gives an outlook on future research.

## 2. Literature Study

### 2.1. Sensor Configuration

The use of DFOS in concrete structures is becoming more and more common. Bado et al. [3] distinguished between laboratory applications, monitoring of buildings, bridges and roads, geotechnical applications, tunnels, pipelines, wind turbines, as well as future applications. All applications have in common that the Rayleigh OFDR measurement principles have recently been increasingly used. A common measurement system is the ODiSI 6100 from Luna Inc., Roanoke, VA, USA It supports strain and temperature measurements. The sensor is the fibre, which consists of the connector for the measuring system, often designed as a so-called pigtail, a measuring fibre and a sensor termination. The components are connected by splicing, the termination can be carried out by a coreless fibre, tight winding (e.g., Ref. [6]) or by cutting the measuring fibre at a 45° angle and placing it in a water bath [7] or epoxy resin [8]. Before the sensor can be used for measurement purposes, it is essential that it is properly configured. The configuration process involves the selection of the fibre type, adhesive, and host material (as depicted in Figure 1, middle). This selection is contingent upon the anticipated mechanical influences (Figure 1, top left), the geometric boundary conditions (Figure 1, top right), the tasks to be analysed (Figure 1, bottom right), and the anticipated chemical influences (Figure 1, bottom left). The research findings regarding Figure 1 are presented and condensed in the subsequent sections.

### 2.2. Fibre Type

The measuring fibre consists of a glass fibre core with cladding, which is surrounded by a coating and optionally other protective layers. The coatings can be stripped mechanically, e.g., PVC or acrylate, or chemically, e.g., polyimide. Figure 2a shows an acrylate coated fibre, Figure 2b a polyimide coated fibre. In Figure 2c a cable with a metal protection layer around the core is depicted. The profiled polyamide layer achieves bonding with concrete.

The fibre types differ in their chemical and mechanical properties. The former can be described, for example, by the resistance to external influences. To investigate these properties, two different sensor cables were considered in Alj et al. (2019) [9]. The first cable has a soft olefin elastomer as an outer layer, the second cable has an outer layer of polyethylene (PE). Both cables were embedded in concrete cylinders and exposed to an alkaline solution (KOH + NaOH, pH = 13.5) for three months to simulate the concrete milieu. Before and after ageing, pull-out tests were performed at different specimen temperatures. The results of the pull-out tests indicated that the first cable exhibited minimal disruption in the bond, whereas the PE cable demonstrated a deterioration in the bond strength following ageing. Bremer et al. [10] carried out investigations on fibres with polyimide, acrylate and carbon coatings. They refer to earlier research that already investigated the improved resistance of bare glass fibres to the alkalinity of concrete through coatings of e.g., polyetherimide [11] or carbon [12]. The fibres used by Bremer et al. [10] were woven into the carbon reinforcement of carbon concrete. It was shown that the alkaline environment had the strongest influence on the polyimide fibres, medium influence on the acrylate fibres and the least influence on the carbon fibres. However, all fibres were damaged, which indicates a poor long-term resistance to the alkaline environment.

The mechanical properties are best distinguished by the structure of the fibre, as shown in Figure 2. Next to the thickness the number of sheaths and layers as well as their bond to the core of the fibre are the main differences. In addition to the moderately thin fibres with coating and the layered fibres, monolithic fibre optic cables have recently been introduced. These were showcased in [13] and studied in [14] for comparison with other fibres in terms of their suitability for crack detection and measurement of crack width.

### 2.3. Adhesives and Host Materials

The choice of adhesive for attachment to the host material also plays a crucial role. Her and Huang [15] presented an analytical model for the transmission of shear forces between composite surfaces. For this, linear elastic isotropic material behaviour was assumed. Also, perfect bonding is assumed and only shear stresses act. It should be noted that only one composite layer was considered here. They found that measured strains are smaller than those actually prevailing on the host structure (here: aluminium). In addition, it was found that with increasing length of the bond and increasing stiffness of the cladding, smaller strain losses were observed. Kim et al. [16] also developed a slightly different analytical model. The aluminum sheathed fibres were bonded to carbon composites. It was found that the strain loss also increases with increasing thickness of the cladding. Alj et al. [17] addressed the composite issue and the influence of adhesive stiffness in crack monitoring on concrete surfaces numerically and with experiments. Fibres were glued into grooves with five different adhesives with Young’s modulus from 0.7 to 11,200 N/mm^2^. These were two-component (2k) epoxy systems and silicone. The study revealed that the softer the adhesive, the lower the strain transfer.

There are rigid adhesives, such as cyanoacrylates and 2k epoxy systems. Silicones are among the softer systems. Barrias et al. [18] used these three types of adhesives to bond polyimide fibres onto reinforced concrete. They have previously noted that strain transfer increases with the stiffness of the adhesive. A softer bond additionally bridges cracks and protects the fibre from mechanical damage. The ideal stiffness of the bond is a compromise between sensitivity and strain. Bado et al. [19] took up the previous approaches of protecting the bond and attached the fibres to reinforcing steel with cyanoacrylate. For this purpose, a groove was made in the steel and the bonded fibre was sealed with a silicone. Previous errors in the measurements were attributed to strain reading anomalies (SRA). The origin of these errors comes from friction between the fibre coating with the concrete. By decoupling with the silicone protectors, the SRAs could be reduced [19]. Galkovski et al. [7] further investigated these grooved rods. A distinction was made between cutting and planing, with the latter being deemed to be more gentle to the steel structure, and thus recommended. Similarly, it was noted that both cold-rolled steel and tempered steel are used for reinforcement. The former has uneven strain curves, which can be attributed to plastic deformations during rolling and thus does not allow a clear strain statement. The problem can be circumvented by applying several fibres. Quenched and tempered steels can loose strength when the grooves are made [7].

Another potential configuration option is the integration of the DFOS sensor into tendons. Gong and Feng [20] embedded a cable into a tendon duct, while Okubo et al. and Oikawa et al. showcase the SmART Strand, a tendon containing an integrated fibre optic sensor within a strand [21,22]. The sensor is enclosed by a plastic sheath. The Smart Tendon research project is pursuing a similar alternative with promising results [23]. The aforementioned methodologies are efficacious for inferring the prevailing prestress through the measured strains, utilising both the SmART Strand and the Smart Tendon.

### 2.4. Conclusions of DFOS Sensor Configuration

Many different fibres with their coatings and cable designs, adhesives and also host materials for the fibres and cables have already been investigated:Host materials:–Concrete surface (with or without groove) or embedded [2,7,10,13,14,17,18,24,25,26,27,28,29]–(embedded) (reinforcing) Steel [7,8,19,29,30,31,32]–Aluminium [15]–Composite [2,16]–Carbon fibre [10]–Tendon [20,21,22,23]Adhesives:–(two component) Epoxy [2,7,8,16,17,18,29,30]–Cyanoacrylate [16,18,19,24,31,32]–Polymere based (e.g., polyester, silicone) [17,18]–Hybride [8]–mineral based (e.g., mortar) [14,24,28,29]–Silicone as protective cover [19,31,32]Fibre cladding/coating or cables:–Acrylate [7,8,10,14,15,29,30]–Polymers (e.g., Polyamide (PA), Polyvinyl chloride (PVC)) [7,15,17,19,29,32,33]–Polyimide [7,8,10,18,24,30,32]–Nylon [14,24]–Other [10,13,14,16,27,29]

The literature study showed that ducts have not yet been used as host materials for DFOS. Furthermore, there is a lack of chemical-based studies on the use of DFOS on rebars.

### 2.5. Validation

In order to ascertain the strains in close proximity to the structure, it is necessary to position the fibre as close as possible to the significant strains of the structure. In the examples shown, tensile tests, compression tests, shear tests or bending tests were carried out. Different methods are used to check the plausibility of the strains measured with DFOS. A comparison with strain gauges is made in [1,2,6,7,15,16,18,24,30]. The tensile forces exerted on the specimen are transferred to a foil situated over the length of the gauge. The result is discrete, averaged to the centre of the strip. Extensometers were used with similar precision in [9,17,19,32] measuring a change of length converted to strain. Digital Image Correlation (DIC) has proven to be more accurate and better comparable for plausibility. This method enables the measurement of two-dimensional or three-dimensional relative displacement of pixels within an image [7,8,19,29,30,32]. It can be observed that there are numerous methodologies for the measurement of strain with DFOS and methods of validation. The consideration of measurement errors (bond, ageing of sensor and measuring device) should not be neglected [34], but is beyond the scope of this paper.

## 3. Embedding Fibres into Concrete Structures

To measure the strains in reinforced concrete, a host material is needed for the fibres to survive the harsh environment. For thin fibres, (acrylate/polyimide coated fibres) rebars are the choice, as they are robust, relatively cheap, available, easy to work with and are used by default in reinforced concrete. Moreover, in reinforced concrete, the tensile forces are absorbed by the rebars as planned due to the low tensile strength of the concrete. They are mainly used in a laboratory environment. Field tests conducted on a bridge in Germany [35,36] revealed that despite extensive protective measures for the sensors and pigtails on the construction site, the majority of the sensors performed poor. Other robust components to install in reinforced concrete or prestressed concrete structures are tendons with ducts and strands. However, strands are likely to be of limited suitability as host material for the fibre, as their standard strain at failure is much lower when compared to steel, and thus the fibre could already be damaged during prestressing or initial loading. In addition, the stranded design of the tendon makes it difficult to neatly attach the fibre. Gong and Feng [20] used special DFOS cables to find grouting defects of tendons. They have shown that DFOS cables can also be installed in tendons. In general, glass fibres can also be installed in cables, which was already shown by some manufacturers. In some cases, multi-layered fibre optic cables are used for this purpose (Figure 2c). The cable shown was used, for example, in [7,24,32] for strain measurement in concrete. The high degree of slip of the fibre due to its structure is a disadvantage of the cable. Another multi-layered cable is used in Bassil et al. [25] and presented in Alj et al. [26]. In the former, the cable is investigated for suitability for measuring crack widths and bonding. In the latter, various properties such as durability and chemical resistance are investigated when the cable is bonded to and embedded into concrete. Among other things, the creep behaviour has been determined.

Monolithic fibre optic cables reduce slip. Herbers et al. [14] compared the ability of five different fibres and cables to detect and quantify cracks in their width. These multi-layered and monolithic cables as well as fibres were bonded into a groove in concrete and onto the concrete surface. A conclusion of the work was the recognition of the good balance between sensitive strain measurement and robustness. The monolithic cable has already been presented for use in concrete structures by Bednarski et al. [13]. Laboratory tests demonstrated its robustness and suitability for crack monitoring. The cables were fixed to the reinforcement before concreting with cable ties. Cables have also been installed in bridge slabs to detect shrinkage cracks. The detection and monitoring of cracks is currently one of the main applications of DFOS in reinforced concrete. As methods improve, more and more accurate information can be obtained. Howiacki et al. [27] introduced the ‘crack shape coefficient’ for crack detection and calibrating different fibres and cables for crack width monitoring.

In Figure 3, the multi layered Solifos V9 cable and the monolithic Nerve EpsilonSensor are shown in a reinforcement cage before pouring the concrete for a test. The fibres originate from geotechnical engineering, were designed for rough conditions in concrete structures and can be used flexibly. They can be purchased prefabricated from the contributor. In this paper, however, we deliberately work with fibres and not with cables. Figure 3 displays fibres coated with acrylate, polyimide and nylon, bonded to reinforcing bars using cyanoacrylate and 2k epoxy adhesive. In Section 5, resistance to diverse chemical influences is tested for the first two variants.

## 4. Use of Ducts as Host Material

### 4.1. Tendons and Their Role in Prestressed Concrete

In reinforced concrete components such as bridges, ceilings or beams, bending moments and thus deformations of the supporting structure occur due to traffic and dead weight loads. Resulting bending moments cause compressive and tensile stresses. Former is primarily transferred by the concrete. Due to its low tensile strength concrete is not well suited to absorb tensile forces. Should the tensile strain in the concrete exceed a certain threshold, it will result in cracking. Now the reinforcing steel absorbs the released tensile forces. However, cracks have a great impact on serviceability and durability. They increase the risk of corrosion, fatigue and deformation. In order to minimise deformation and thus cracking of the structure, as well as to increase the possible span widths, a tensioning of the structure is generated by prestressing. In the case of non-centric tension, shown as red line in Figure 4 as $P_{e\;\neq \;0}$), a bending moment $M_{y}$ is induced by the eccentricity *e* to the neutral axis. No bending moment results by centric pretension, shown in Figure 4 as a blue line ($P_{e=0}$). The prestressed steel attempts to revert to its original state, thereby opposing any deformation. The structure now has a residual stress state with compression in the concrete and reinforcement without cracks and deformations.

A common prestressing technique for prestressed concrete bridges is post-tensioning with subsequent bond. For this purpose, tendon ducts made of sheet steel or plastic are placed in the formwork. The strands are either inserted into the duct before or after the concrete is poured. After the curing of the concrete, the strands are tensioned against the concrete with hydraulic cylinders. The tubes are then grouted airtight, for example with mortar, creating a bond between the strand and the duct.

### 4.2. Basic Idea and Specimens

In addition to the installation of cables, the use of tendons, and the bonding of fibres to reinforcement, the application of fibres to ducts is presented in this chapter. We examine the specimen introduced by Otterbach [37]. He tested an application that is particularly intended to map loss of the post-tensioning force. For this purpose, the application of DFOS to ducts was investigated using two different methods:Attach DFOS in longitudinal direction (Figure 5a);Attach DFOS radial (Figure 5b).

The application of the fibres with polyimide and acrylate coating was achieved through the use of a cyanoacrylate adhesive. Even with a step-by-step application with tightening of the fibre by adhesive tape and hardening of the adhesive, a clean bond was not possible. For this reason, this variant was not considered further and only variant number two was pursued although it is not ideal either. A long sensor length is required to cover a comparatively small section of the tendon. In addition, the tight bending radii are a possible source of SRAs. Figure 5c shows a sketch of the duct with the fibre paths. The weld is shown as a grey area. It can be seen that the weld runs longitudinal of the actual duct (Figure 5b), whereas in the sketch the weld is radial ((Figure 5c). Unfortunately, no ducts like the one in Figure 5c could be examined.

### 4.3. Theory

With a host material, the concrete strains can be measured directly at the tendon, whose change in strain is equal to the concrete ones. Consequently, the duct is subjected to overcompression, while the prestressing steel is responsible for absorbing the tensile stress. The corresponding strain on the duct depends on a number of factors, including the load, the position of the tendon (*e* in Figure 4), and the time-dependent prestressing force of the tendon. In EN 1992-2 [39], the decompression check must be carried out for an application in bridge structures for prestressed concrete components. This implies that the tendons must be situated in the region of the structure that is experiencing pressure. The compression of the concrete should be a maximum of $\varepsilon_{c}\approx 0.45\times \left(f_{\mathrm{cm}}/E_{\mathrm{cm}}\right)$, which is for a concrete class of C30/37 [40] $\varepsilon_{c}\approx 0.52‰=520$ µm/m. The strain decreases rapidly under load. During decompression, $\varepsilon_{c}$ experiences tensile strain in the area of the tendon.

If the fibres are attached to the coil, the following theoretical approach to strain applies. In the range of a pitch (Figure 5c), a fibre not only measures the length of the pitch $l_{\mathrm{pitch}}$, but also once the circumference of the duct with the nominal diameter $\phi_{\mathrm{nominal}}$ (Figure 5c). Therefore, the measured strain in the fibre $\varepsilon_{\mathrm{fibre}}$ is by the factor $k_{\mathrm{pitch}}$ lower than the actual strain in the duct $\varepsilon_{\mathrm{duct}}$.
(1)$$\varepsilon_{\mathrm{fibre}}=k_{\mathrm{pitch}}\times \varepsilon_{\mathrm{duct}}$$

The factor $k_{\mathrm{pitch}}$ results from the ratio of $l_{\mathrm{pitch}}$ and the total length $l_{\mathrm{tot}}$.
(2)$$k_{\mathrm{pitch}}=\frac{l_{\mathrm{pitch}}}{l_{\mathrm{tot}}}$$
where:(3)$$l_{\mathrm{tot}}=l_{\mathrm{pitch}}+\phi_{\mathrm{nominal}}\times \pi$$

The parameters $l_{\mathrm{pitch}}$ and $\phi_{\mathrm{nominal}}$ are predefined for each duct. $k_{\mathrm{pitch}}$ can vary greatly. Using the example of a ducts series from the company SUSPA DSI [38] in Table 2, it can be seen that the factors can vary from approx. 0.171 to approx. 0.060.

The LUNA ODiSI interrogator acquires the measurement data with a resolution of 0.1µm/m. A change of 0.1 µm/m would cause a change of $\Delta \varepsilon_{\mathrm{duct}}=\frac{\Delta \varepsilon_{\mathrm{fibre}}}{k_{\mathrm{pitch}}}=\frac{0.1\;\mathrm{µm}/m}{0.131}=0.76$ µm/m for a duct with $\phi_{\mathrm{nominal}}=55\;\mathrm{mm}$. Regarding this the resolution of the application should be sufficient to draw conclusions regarding the condition of the tendon.

### 4.4. Test

Due to its inherent stability, higher loads can be achieved with tensile tests than with compression tests. The duct used is a SUSPA Strand DW duct for post-tensioning of structural components with seven strands (ϕ=15.7mm each) from DYWIDAG-Systems International GmbH and is subject to ETA 13/0839 [41]. Key information of the duct: $\phi_{\mathrm{nominal}}=57\;\mathrm{mm}$, $l_{\mathrm{pitch}}=19\;\mathrm{mm}$, height of the profiling h=3mm. However, the geometry and the material properties may vary slightly depending on the batch and manufacturer. The duct was made of cold-rolled strip steel DC01 MA LC according to DIN EN 10139 [42] with a width of 36mm and a sheet thickness of t=0.25mm. The cross-sectional area is therefore
$$A\approx \phi_{\mathrm{nominal}}\times \pi \times t=57\;\mathrm{mm}\times \pi \times 0.25\;\mathrm{mm}\approx 45\;{\mathrm{mm}}^{2}.$$
with the materials roughly estimated yield strength of $R_{e}=140\;{N/\mathrm{mm}}^{2}$ the characteristic resistance of the duct is estimated as
$$F_{R,k}=A\times R_{e}=45\;{\mathrm{mm}}^{2}\times 140\;{N/\mathrm{mm}}^{2}=6300\;N.$$

At a load up to two thirds of the yield strength ($\sigma =0.67\times R_{e}\approx 93\;{N/\mathrm{mm}}^{2}$, $F_{0.67\mathrm{Rk}}=4.2$ kN), according to Hooke’s law, the duct is stretched to
εduct,0.67Re=σE=93N/mm2210,000N/mm2≈444µm/m
assuming a modulus of elasticity of $E=210,000\;N/{\mathrm{mm}}^{2}$. The expected strain in the fibre should be according to Equations (Equation 1)–(Equation 3) approximately
$$\varepsilon_{\mathrm{fibre}}=k_{\mathrm{pitch}}\times \varepsilon_{duct,0.67Re}=0.096\times 444\;\mathrm{µm}/m=43\;\mathrm{µm}/m$$
with $k_{\mathrm{pitch}}=\frac{19\;\mathrm{mm}}{198\;\mathrm{mm}}\approx 0.096$ and $l_{\mathrm{tot}}=19\;\mathrm{mm}+57\;\mathrm{mm}\times \pi \;\approx 198\;\mathrm{mm}$.

The duct to be tested was equipped with a polyimide fibre using a cyanoacrylate adhesive. The sensor was reset (tared) before the tensile force was applied. The test was performed with a Zwick Z400 universal testing machine (Figure 6a). The clamping jaws to support the pulling of the duct are shown in red colour. Figure 6b shows the applied load as a function of time. The load was increased in 0.5 kN increments up to 4 kN from a starting load of 1 kN and held for approximately 30 s to obtain sufficient data per load increment. The sampling rate was 50 Hz and the gauge length was 2.6 mm. No additional sensors were installed as this was a feasibility study.

### 4.5. Results

A series of pre-processing steps were employed in order to reduce the data size. The ‘segment of interest’ (Richter et al. [43]) excludes sensor parts not connected to the duct, namely the beginning (pigtail, 0.9 m) and termination (approximately 0.1 m). Roughly 31 % of the data could be eliminated without losing essential information. The measurement data recorded also consists of many SRAs, where measured values either appear as Not a Number (NaN) or in an order of magnitude that is represented in the measurement results as unrealistic strain peaks ($\varepsilon \approx 10^{8}$) and thus are eliminated.

In addition, timestamps at which the sensor did not provide a response to the measuring device are also removed with a tolerance. This results in cleaned data (Figure 7b) from the raw data (Figure 7a). However, there is a slight amount of noise in this measurement data [14]. Therefore, a moving average across a sliding window of ±15 measurements was implemented leading to filling of the gaps in the data (Figure 7c). The analyses were conducted using the data smoothed over time.

### 4.6. Evaluation

The processed data in Figure 7c shows a significant reduction in the cleaned measurement data. Despite the poor data quality, an oscillating pattern can be observed. The period of this pattern falls within the range of 190–210 mm, which corresponds to the theoretical circumference and pitch of the duct of 198 mm. This variation in strain is non-uniform across the entire cross-section. One possible reason for this is an eccentric load application from imperfections in the rigid mounting to the testing machine of the friction-based clamping device. This led to plastic deformation in the mounting area of the duct. The load conditions do not align with those in a concrete structure. The force is transferred to the duct by bonding with the strand and the surrounding concrete.

Another factor contributing to the insufficient data quality is the fibre application process to the duct. The fibre is bonded to the duct in several stages, with half of its circumference glued from above to ensure a secure hold. This measure is necessary because the fibre would not adhere to the adhesive otherwise. Unfortunately, this process exposes the adhesive to varying environmental conditions, including the adhesive’s curing time, temperature, and humidity levels, resulting in inconsistent curing conditions and variable bonding conditions. Additionally, without a guide for the fibre on the duct, geometric errors may occur. This can be classified as a manufacturing error.

The measured strains diverge significantly from the analytically calculated values. One reason for this can be a bending moment due to the eccentric load. Another cause could be the impact of horizontal strain, as fibres only measure in the longitudinal direction and therefore no transverse strain. Drake et al. [44] conducted tests to measure strains on a rectangular plate. DFOS were applied longitudinally, transversely, and diagonally. The measured strains were independent of each other, as the Poisson’s effect was prevalent in the sample. Winkler et al. [28] investigated the influence of fibre orientation on crack width, observing the occurrence of deviating strains at varying angles. Other effects like residual stresses introduced during rolling or deformations may affect the duct. Due to the tare before the measurement, the influence of residual stresses can be reduced.

## 5. Exposure to Environmental Conditions

### 5.1. Environment in a Concrete Lifecycle

The sensor attached to the rebar is exposed to different conditions and pH values throughout its operational lifespan. After manufacturing, the sensor rod is stored in a dry place and experiences normal seasonal and regional humidity and temperature. During storage, transport, and installation, the rod may be exposed to mechanical stresses, which are not considered here. During that processes, the rods are exposed to weather conditions until concreting. This includes exposure to ultraviolet radiation and water from rain or standing water, which has low mineral content, although it can become contaminated.

While pouring, the steel is once again exposed to mechanical stresses because of falling concrete and compaction. The alkaline environment, combined with the prevailing fresh concrete pressure, also has a chemical effect on the bars. The reinforcing steel is not only resistant to alkalinity but is also protected against corrosion (passivity).

After the concrete has cured, no mechanical effects except for creep, shrinkage and strains in the steel resulting from external influences are expected during the regular operation of the structure. As the concrete ages and is exposed to chloride, the reinforcing steel may become depassivated, losing its protection against corrosion and leading to its deterioration. After categorising the structure into an exposure class according to DIN EN 206 [45], the rebar is protected by a concrete cover. For bridge structures, typical classes include XC4 for external components from carbonation and XD1 or XD2 from deicing salt (depending on the component) for moderately moist or wet components.

### 5.2. Basic Idea and Specimens

The presented conditions in Section 5.1 are described as environmental exposures.

Storage in air (LL)Storage in water (WL)Storage in alkaline environment (BL)Storage in saline (SL)

Previous experiments focused primarily on mechanical influences, such as [7,13,14,18,19]. The sensors were developed to withstand the extreme effects of concreting. Several methods were presented in Section 2.4 for attaching the fibre to the rebar. The literature research demonstrates that good protection can be achieved by creating a groove in the area of the longitudinal rib. Sealing the groove after gluing in the fibre is also effective. Tests were conducted by Müller [46] to investigate the durability of the fiber optic sensors. For this purpose, rebars with a longitudinal groove that was approximately 49 cm in length were prepared. Figure 8b shows their profile. The sensor configurations each consist of a fibre and an adhesive. The fibre types used have polyimide coating and acrylate coating. The adhesives used were a cyanoacrylate and a two-component epoxy resin. The protective silicone layer was not applied as it could have affected the test results. The behaviour of sensor rods, fitted with sensors in different environments, was tested. The air temperature for LL was approximately 20 °C with a relative humidity of 40–50%. Distilled water at approximately 18 °C was used for WL. For BL, a solution of NaOH, KOH and water with a pH value of 13 was used. For SL, a slightly acidic solution (pH = 5) with NaCl was used. Figure 8a provides an overview of the resulting sample definitions. A total of three samples per configuration, four storage milieus, two types of fibre, and two adhesive types result in a total sample number of N=4×2×2×3=48.

### 5.3. Theory

The rebars used as host material contribute to the load-bearing capacity of the structure when embedded in concrete. They experience the same strains as the load-bearing reinforcement and therefore the increase of strains while cracking is detectable. In the service condition, the strains are in an elastic state, i.e., in the range before the steel yields. The corresponding strain to the yield point $R_{e}$ is equal to $\varepsilon_{e}\approx 2500\;\mathrm{µm}/m$.

The tensile test load is within the elastic range using the testing machine described in Section 4.4. The first strain measurement T01 is taken. The tensile test is then repeated (T02) after storage in the respective environment. Any deviations in the measured strains should be analysed to draw conclusions about the quality and practicability of the sensor configuration. The design incorporates a groove to reduce the cross-section. As the samples were all made by hand, it is possible that there may be irregularities in the depth and straightness of the groove. These factors are considered when analysing the test results.

### 5.4. Test

The samples were produced from a single batch of reinforcing steel with grade B500A according to DIN 488-1 [47], DIN 488-2 [48] and EN 1992-1-1 [40]. The bars have a nominal diameter of d=12mm, resulting in a cross-sectional area of $A_{s}=113\;{\mathrm{mm}}^{2}$. According to the standard, the bars possess a characteristic yield strength of $f_{\mathrm{yk}}=500\;{N/\mathrm{mm}}^{2}$ and a Young’s modulus of approx. $E\approx 200,000\;{N/\mathrm{mm}}^{2}$. This results in a characteristic limit load of approx. $F_{\mathrm{yk}}=f_{y,k}\times A_{S}=500\;{N/\mathrm{mm}}^{2}\times 113\;{\mathrm{mm}}^{2}\approx 56\;\mathrm{kN}$. To ensure safe testing within the elastic range, the limit load of the centric tensile test is set at $F_{\lim}=0.5\times F_{\mathrm{yk}}=28\;\mathrm{kN}$. The estimated strain is about $\varepsilon_{\lim}=\frac{500\;{N/\mathrm{mm}}^{2}}{2\times 200,000\;{N/\mathrm{mm}}^{2}}=1250\;\mathrm{µm}/m$.

The load in the testing machine is increased with a constant loading speed of 0.3 kN/s, held for 60 s and released at 0.3 kN/s to its initial position (see Figure 9b). The strain is measured using the LUNA ODiSI interrogator with a gauge length of 0.26 mm and a sampling rate of approximately 5–9 Hz. Prior to commencing the test, the sample is clamped and the measured values are tared. The strain reference is a LIMESS Real Time Strain Sensor (RTSS) video extensometer with marks on the rebar (Figure 9c). Prior to each test, the RTSS measurement was calibrated. The measured values represent the average strain between the two marks. The test is load controlled, using the Zwick Z400’s load cell.

After T01 the samples were divided into four equal quantities and stored for 28–30 days in four distinct storage environments. The storage conditions were monitored to ensure consistency. Following the storage period, the specimens were dry cleaned to remove any corrosion and storage fluids, ensuring better fixture in the testing machine’s clamping system and avoiding contamination. The specimens were reconnected to the measuring device. After storage in the milieus, five sensor configurations failed. The fibre optic measuring device was tared again before conducting the test T02 on the remaining specimens with the load curve shown in Figure 9b. The results of the two tensile tests and the failed specimens are presented in the following section.

Without a quantitative measurement, no statement about the impact on the sensor during storage due to the second tare is possible. The strain values are stored in a JSON-based file over the sensor length at the time of the tare. This information can be expressed by the vector ${\vec{s}}_{\mathrm{tare}2}$. The application of this tare results in the original calibrated strain measurement ${\vec{s}}_{k,i}$ for each time stamp *i* being subtracted by the stored values, thereby producing the modified vector ${\vec{s}}_{T02,i}$ for each time point *i*. The strain values of the second tare are then added to return the values to the original plane ${\vec{s}}_{k,i}$. The subtraction of the values of the initial tare, designated as ${\vec{s}}_{\mathrm{tare}1}$, allows a comparison with the T01 strain measurement. This is because the measured values are once again at the level of the initial tare in the vector ${\vec{s}}_{T01,i}$ (Equation (Equation 4)).
(4)$${\vec{s}}_{T01,i}={\vec{s}}_{T02,i}+{\vec{s}}_{\mathrm{tare}2}-{\vec{s}}_{\mathrm{tare}1}$$

### 5.5. Results

#### 5.5.1. Pre-Processing of DFOS Data

The specimens have different sensor lengths, the two measurements of a specimen T01 and T02 have identical lengths. Furthermore, the modified T02_T measurement data is calculated using Equation (Equation 4), which enables quantitative comparison with the T01 measurements. To enhance comparability between samples and to reduce measurement data, data is cut to the segment of interest. The sensor rod’s labelled gauges are used to delimit the area at the beginning, and data frames are cut out in this region (see Figure 10a). For each measurement, T01 is plotted at the time stamp *t* = 110 s, as the load increase of the test is F’ = 0 kN/s and the strain is therefore almost constant. A segment of interest of approximately 25 cm is selected, starting 5 cm below and ending 5 cm above the marked area in Figure 9c.

The period of interest is determined using the load curve over time, as shown in Figure 9b. It corresponds to the time during which the load remains constant at F = 28 kN, with a 10 s exclusion on each side of the time line to eliminate any time offsets when starting the measurements. This results in a period of interest ranging from 110 s to 150 s. Figure 10c displays the resulting unprocessed data. The strain values vary over time and sensor length, indicating a need for further investigation. Various processes are available for this purpose, but their use depends on how the data is utilised. To evaluate the quality of the combinations, no SRAs or dropouts need to be removed in this case, as shown in [43]. However, smoothing the curves can be useful for noise reduction. The recorded measurement data can be smoothed in two directions: by time and by length. In Figure 10d,e, the measurement data was processed using a floating average mean of ±5 around the respective measured value. Three different variants were distinguished.

Gauge smoothed: The data obtained from measurements is smoothed in the direction of the sensor using gauges. This procedure ensures that the smoothed values of a measurement point in Figure 10d can deviate from the actual values (original) over time without eliminating large peaks. In a section taken at a specific point in time, the curve is significantly smoothed (see Figure 10e).Time smoothed: The data is smoothed over time using the timestamps. This process significantly smooths the curve when intersecting a gauge in Figure 10d. When intersecting a timestamp in Figure 10e, the deviation from the original is minimal.Gauge and time smoothed: If both smoothing variants are executed together, the result is smoothing oriented in the time and sensor directions. The order of operations does not affect the results for the data sets used in this study.

The use of gauge smoothing is not recommended for the measurement data analysed in this chapter as it removes important local information, such as stress peaks that indicate poor sensor configurations. Instead, time smoothing is more appropriate. However, it is important to investigate whether the temporal fluctuations in the strains are due to uneven load application. Figure 11 displays the strains in a section through a gauge compared to the forces in the specimen over the test period. The forces oscillate with an amplitude of up to 0.01 kN. The amplitude decreases over time, and so does the frequency of the load change. This behaviour is possibly due to the force-controlled test set to 28 kN and can be interpreted by a slight slipping of the specimen. The ‘Force in specimen smoothed (10)’ curve and the ‘Strain in specimen smoothed (10)’ curve are both moving averages of the values, with a window of ±10 values. Similarly, the ‘Strain in specimen smoothed (5)’ curve is a moving average of the values, with a window of ±5 values.

Upon comparing the two smoothed curves of the strains, it is evident that the peaks and valleys of the finer curve are in closer proximity to the raw curve of the force. When comparing the curves of strain and force smoothed with a moving average of 10, it can be observed that the curves are only partially synchronised from 110 s to approximately 134 s. From 134 s to 150 s, the two curves are almost symmetrical. Upon comparison of the two amplitudes, it is evident that they correspond well in quantitative terms. This can be demonstrated by comparing the valley at 137.8 s to the peak at 140.6 s. To achieve this, the percentage increase in force and strain from $t_{1}=137.8\;s$ to $t_{2}=140.6\;s$ is compared.
$$\Delta F_{\mathrm{rel}}=\frac{F_{\mathrm{smooth}}\left(t_{2}\right)}{F_{\mathrm{smooth}}\left(t_{1}\right)}=\frac{27.994\;\mathrm{kN}}{27.988\;\mathrm{kN}}=1.00021$$
$$\Delta F_{\mathrm{rel}}=\frac{\varepsilon_{\mathrm{smooth}}\left(t_{2}\right)}{\varepsilon_{\mathrm{smooth}}\left(t_{1}\right)}=\frac{1252.39\;\mathrm{µm}/m}{1252.01\;\mathrm{µm}/m}=1.00030$$

The increase is within a similar range. Based on these brief analyses, it is evident that a moving average of 5 is more accurate than a moving average of 10. Additionally, when compared to a moving average of 7 and 15, a range of 5 also seems to be appropriate for smoothing around each value. Furthermore, it is apparent that the period between 134 s and 150 s is more suitable for evaluation than the initial range. As a result, all subsequent quantitative analyses will be conducted with these processing.

The last step in data pre-processing is to transform it into a standardised timeline. Each specimen is subjected to the same load curve, albeit with slight variations due to load scattering and different test start times. The raw measurement data timestamps are displayed in a timeline format of date:time. This means that comparative operations between tests cannot be performed. The tests are first converted into seconds before the initial operations shown here. After adjusting the measured values, the load curves are superimposed to match the strain curves. Due to the possibly different sampling rate of the fibre optic measurement, this results in an offset of the time stamps. These points, which are within a few seconds range, are then synchronised. The time stamps of the data frames for tests T01 and T02 are merged, and any empty positions that do not exist in the respective test data are filled by interpolation to the nearest measured values. This generates additional supporting points of the linearly connected measuring points, which are fictitious sensor values that do not falsify the results.

#### 5.5.2. Qualitative Results

After pre-processing, the strain curves were then plotted, as shown in Figure 10c. A distinction was made between the sample name and the test times T01 and T02. The sample name is important to determine which fibre/adhesive environmental combination has an influence on the results. The test times are important to determine if the sensors showed damage before or during the first test T01. These curves can then be qualitatively assessed into categories 0 to 4. Categories 0 and 1 are treated together as their results are similar. Figure 12 provides examples of the categorisation of the pre-processed results, which are explained below.

Category 1: No measurement values available: It is important to distinguish whether no measurement data could be collected in both test runs or only in the second one. If no measurement data was collected in one run, it may indicate a faulty sensor rod. This could be due to damage caused during fabrication or during storage. As a result, the sensors no longer have a connection to the measuring device and are no longer recognised as sensors. These specimens are classified as subcategory 0. Category 1 includes all specimens recognised by the measuring device that only consist of dropouts and SRAs in the data of interest. It is depicted in Figure 12 (Cat. 1).Category 2: Measurement values with significant errors: This category contains data with significant anomalies, such as pronounced peaks and many dropouts. An example of this can be seen in Figure 12 (Cat. 2). The left graph clearly shows the strong peaks and the sensor failure after a certain time. In the right graph, the sensor failure can be seen in a medium time segment and SRAs in working segments. Loose contacts in the pigtail or splice points can cause issues with the measuring system’s ability to accurately record backscatter and transform it into strain.Category 3: Slightly erroneous measured values present: The measured values contain few anomalies, mostly SRAs or dropouts that are isolated in terms of time or location. This is illustrated in Figure 12 (Cat. 3) with two examples.Category 4: No abnormalities: In this case, the analysed measured values are free of errors. This is demonstrated in Figure 12 (Cat. 4). However, the quality of the curve, based on quantitative characteristics, such as noise or plausibility of the measured strains, has not yet been evaluated.

Figure 13 shows the categorised measured values of all tests. The table’s structure is used frequently in the following evaluations. It is divided horizontally into two blocks: Test T01 and test T02. The vertical separation is by category. The combinations of fibre and adhesive are shown from left to right. Samples 01 to 03 are always displayed vertically. In the upper half (T02), the storage conditions prior to T02 are separated by dashed lines. This could also be applied to the lower block T01, although it is irrelevant to the objective evaluation of the samples T01 as to which storage condition they are later exposed to. By superimposing the two blocks one can observe the change in the respective specimen.

Mathematical operations should be performed on the measurement data for further analysis. Figure 13 identifies measurements with faulty values. For comparability, both measurements (T01 and T02) must have reasonably good values. Test specimens with at least one measurement in Category 0 or 1 are no longer considered for further investigations, so are most measures in category 2. Only sample LLpolEP01 is an exception to this rule due to the presence of only NaN values and no outliers.

One method of assessing the quality of the measured data is to use regression analysis. The measurement data has a single strain value for each distributed location and time stamp, allowing for regression to be mapped in both the direction of time and sensor length. A function can be created, with the most common being a linear mapping of the available points. For this purpose, one seeks a linear function $y=a_{1}\times x+a_{0}$ that minimises the error between the measurement data points and the curve’s interpolation points. The coefficient of determination $R^{2}$ expresses how well the curve fits the data set. The closer $R^{2}$ to 1.0, the better the fit of the numerical solution to the data set. Polynomial regression analysis can be performed for functions of the n-th degree, such as 2nd degree: $y=a_{2}\times x^{2}+a_{1}\times x+a_{0}$ or 3rd degree: $y=a_{3}\times x^{3}+a_{2}\times x^{2}+a_{1}\times x+a_{0}$. The greater the degree of the function, the greater $R^{2}$, indicating a more accurate fit to the strain curves. This method can be used to predict values through interpolation and extrapolation. However, the use of extrapolation is associated with a high degree of uncertainty, especially for regressions with higher degree polynomials. This is because the function values can increase rapidly with larger x-values. For this particular case, evaluations up to 3rd degree polynomials are sufficient.

The regression models utilised in the results origin from the Python framework SciKit-Learn 1.4.1. [49]. The ‘Linear Regression’ and ‘PolynomialFeatures’ classes are employed. The models also incorporate outliers (e.g., SRAs) but are unable to process NaN. Therefore, NaN values are skipped in the regression analysis. For the qualitative analysis, regression curves up to the 3rd degree polynomial are formed for all measurement series T01, T02, and the transformed measurement data T02_T. Both raw data and smoothed data are analysed, as shown in Figure 14. All measurement location time stamps are used, resulting in the points appearing as bars for each gauge. A regression curve is formed for each time stamp, and the regression curve shown is the mean of all the curves formed. It is evident that the raw data (Figure 14a,c) is more scattered than the smoothed data. However, the two measurements analysed differ not only in their form but also in their data quality. The strain values for the measurement SLacrCA03T02measure_T (shown in Figure 14a,b) exhibit a clear decrease over the sensor length, with only minor fluctuations. There is a noticeable lack of significant scattering. Therefore, a linear regression is visually appropriate, and the coefficient of determination ($R^{2}$ = 0.94) supports this. The polynomial regression curves also have a similar $R^{2}$ value. However, the curve LLpolEP02T02measure (Figure 14c,d) cannot be accurately represented using linear regression ($R^{2}$ = 0.31). A second-degree polynomial regression provides better results in this case. The most accurate result is obtained using a third-degree polynomial regression with an $R^{2}$ of 0.68. However, the $R^{2}$ is not suitable for making predictions and is therefore labelled as ‘bad quality’.

The comparison of the two tests shows that the differences between the raw data and the smoothed data are small, which further emphasises the use of smoothed data for better clarity. In addition, each measurement series has its own characteristic for the $R^{2}$ of the respective regression. Qualitatively, this can primarily be described by which regression curve fits the data set well. The following boundary conditions apply to this paper:If the $R^{2}$ are similar (relative standard deviation of the three $R^{2}$ values less than 5%), then the curve can be seen as a linear regression.If the $R^{2}$ values of the linear regression is significantly smaller than the $R^{2}$ of the polynomial regressions, these are compared. The measured values are categorised as 2nd degree polynomial regression if the following applies: $\frac{R_{\mathrm{pol}3}^{2}}{R_{\mathrm{pol}2}^{2}}\leq 1.05$. Otherwise it is categorised as 3rd degree polynomial regression.The coefficient of determination of the categorisation applies to the categorised curve.In general: A categorisation as ’bad regression’ applies if $R^{2}\leq 0.70$.

Figure 15 shows the result of the qualitative categorisation. As in Figure 13, the samples are assigned fixed positions within a field. One can see that the polyimide-coated fibres exhibit high scattering after storage and cannot be assigned to one of the three regressions. However, the fibres coated with acrylate and bonded with cyanoacrylate (acrCA) show consistent allocation in the regression curves across all storage types.

#### 5.5.3. Quantitative Results

To quantitatively evaluate the measured values for the resistance of fibre/adhesive combinations for strain measurement against the effects, four procedures are examined:Minimum/maximum values of the measured strains.Average value of the measured strains.Stability of the measurement over time using the regression curves.Comparison of the measurements using the $R^{2}$ of the regressions.

The maximum and minimum strains for the smoothed data of all test series are determined over the considered measuring range, creating a corridor within which the measured values for the respective measurement lie. Figure 16 shows these values separately for all tests. The corridor typically has a width of approximately 0–200 μm/m. In a measurement where the force is almost constant, the strain curve should also be almost straight. Although various effects and imperfections may limit the results, there is a slight tendency that the sensor rods with polyimide-coated fibres have the widest corridors.

The corridor for the polyimide-coated fibres bonded with cyanoacrylat (polCA) across all samples, except one, significantly increases between T01 and T02 or T02_T. However, almost no influence can be seen for acrCA. For the fibres bonded with EP, no discernible trend is observed as some corridors increase while others remain the same.

In order to verify the measured strains, one method is to compare the mean strains. The mean strain for each measurement must be calculated, with the exclusion of dropouts and SRAs. As the tare was carried out after the clamping and before the load test, it is not practical to compare T02_T with one of the other two test series. Therefore, only T01 and T02 are under consideration. The difference $\Delta {Ø}_{T01,T02}\;=\;$${Ø}_{T01}\;-\;$${Ø}_{T02}$ is determined for each sample. The left-hand subgraph of Figure 17 shows that the average deviations of the curves are in the range of approximately −80 to +70 μm/m. The deviations were lower (between −20 and +42 μm/m) for the samples of the polyimide-coated fibre with 2k epoxy (polEP). All other samples had at least one outlier. Allowing one outlier, the curves of acrCA and polCA also show acceptable results. The acrylate coated fibres with 2k epoxy bonding (acrEP) configuration has two outliers. The mean values of the T01 measurements are displayed on the right-hand side of Figure 17. The expected measured value is approximately 1250 μm/m (indicated in Figure 17, right), while the measured strains of the samples are mostly higher. The maximum deviation (LLacrEP02T01) is approximately 100 μm/m, which corresponds to approximately 8.2%. The values are slightly higher than the variation of the material characteristics of the reinforcement. The cross-sectional dimensions and modulus of elasticity were not determined.

A more precise method is to quantitatively compare regression curves based on their coefficients of determination. First, the type of curve in Figure 15 is qualitatively categorised, and then the absolute values are considered. Next, the extent to which the regression curves vary over time is analysed by determining the relative standard deviation per measurement for each individual regression curve. The coefficients of the polynomials are used to calculate the mean values and standard deviations. The mean value is then calculated from the relative standard deviations of the individual coefficients. Figure 18 shows an example for linear regression, with the values plotted logarithmically in the x-direction. It is evident that the polCA samples in T02 and T02_T have a higher temporal dispersion compared to measurement T01.

The typical scattering range is approximately 0.05–1% when using linear regression, which is also where most of the measurement curves are located. The acrCA combination only deviates from this range once upwards. The higher the relative standard deviation over time, the less stable the combination against the permanent influence of strain or previous exposure to an environment.

In the next step, the $R^{2}$ of the curves are compared. A vector is created by combining $R^{2}$ of the linear, 2nd, and 3rd degree polynomial regressions in Equation (Equation 5).
(5)$$\vec{R^{2}}=\left(\begin{array}{c} R_{\mathrm{lin}}^{2}\\ R_{\mathrm{pol}2}^{2}\\ R_{\mathrm{pol}3}^{2} \end{array}\right)$$

For all measurements, a point in space can be depicted (Figure 19a). However, this view is not suitable for comparing measurement times. Instead, we select the Euclidean distance between the $R^{2}$ vectors of two measurement times, as demonstrated in Equation (Equation 6) for T01 and T02.
(6)$$\Delta \left({\vec{R^{2}}}_{T01},{\vec{R^{2}}}_{T02}\right)=\sqrt{{\left(R_{\mathrm{lin},T01}^{2}-R_{\mathrm{lin},T02}^{2}\right)}^{2}+{\left(R_{\mathrm{pol}2,T01}^{2}-R_{\mathrm{pol}2,T02}^{2}\right)}^{2}+{\left(R_{\mathrm{pol}3,T01}^{2}-R_{\mathrm{pol}3,T02}^{2}\right)}^{2}}$$

This creates a relationship between the two vectors, resulting in a distance. Figure 19b shows the distances of all T01-T02 and T01-T02_T. It is expected that the coefficients of determination do not differ greatly, as neither the fibre nor the bar has changed. Changes are either due to the adhesive-fibre-steel bond or the clamping conditions. The maximum distance $d_{\max}$ occurs when all individual distances are 1, so $d_{\max}=\sqrt{3}\approx 1.73$.

To improve the categorisation of the results shown in Figure 19b, Figure 20 displays two plots. Each plot shows a sample with three measurement curves, including all the smoothed measurement curves obtained. Additionally, the plot includes three regression curves for each of these curves (linear, polynomial of 2nd degree, polynomial of 3rd degree), with their $R^{2}$ values noted in the legend. Figure 17b highlights the vector distances $\Delta R_{T01,T02}^{2}$ and $\Delta R_{T01,T02\_T}^{2}$. In the example of LLacrCA03 (Figure 20a, red circle in Figure 19b), the vector distances are $\Delta R_{T01,T02}^{2}=0.49$ and $\Delta R_{T01,T02\_T}^{2}=0.01$. The former is evident, because the curve of T02 fits only the 3rd degree polynomial regression well, while T01 delivers a good $R^{2}$ in all three regressions. Nevertheless, an adjustment of the measurement to the tare of T01 (appearance of T02_T) results in a good fit of the curves T01 and T02_T, which is also expressed by $\Delta R_{T01,T02\_T}^{2}$. However, this does not reveal the shape of the curve. It only indicates a fitting regression model with a high $R^{2}$ value can be achieved.

Figure 20b shows that the vector distances for WLacrEP03 (grey circle in Figure 19b) are $\Delta R_{T01,T02}^{2}=0.09$ and $\Delta R_{T01,T02\_T}^{2}=1.58$. Although $\Delta R_{T01,T02}^{2}$ is low, the three curves differ. Both polynomial regressions provide good fits, but only one linear regression does. If the criterion of the linear regression were omitted, the $\Delta R_{T01,T02}^{2}$ would decrease and indicate a match of the curves (with potential shift in y-direction), which would be incorrect. It is evident that none of the three given options provide a correct regression for T02_T when considering $\Delta R_{T01,T02\_T}^{2}$. The corresponding $R^{2}$ values are low, and all attempted regressions result in an increase of $\Delta R_{T01,T02\_T}^{2}$.

### 5.6. Evaluation

In the following section the results are evaluated and presented in Table 3. The qualitative results categorise the measurements for each test specimen based on the quality of the data (see Figure 13). It evaluates the qualitative influence of the storage on the rods. For the acrCA combination, storage in the respective media does not seem to affect the consistency of the measurement data. During the first test of acrEP, all three test specimens for BL storage failed, and the measurement data was only checked after the test, making it impossible to repeat the tests. Additionally, two specimens did not respond after air storage. The quality of the remaining samples was not significantly affected by storage. The quality of the samples with polEP is inconsistent. Only sample BLpolEP01 was not recognized by the measuring device after storage, indicating that the sensor was likely damaged. The most significant total failures were recorded by polCA. Losses in data quality were observed in all storage types except air. In summary, failures occurred in all samples, rendering measured values unusable or unavailable for quantitative analysis. The *Category Score* line awards scores (1–4). Once sorted into categories, there are eight specimens available for further polEP and polCA analyses, seven for acrEP, and 11 for acrCA.

During the quality assessment, we analyse whether the measurement curves fit a linear function or a 2nd or 3rd degree polynomial function through regression analysis (Table 3—*regression type*). Here the test T01 is compared with T02 or T02_T to see whether there are differences in their categorisation. Jumps from ‘regression’ to ‘no regression’ indicate further dispersion of the measurement data over the sensor length, such as sinusoidal progression. This is particularly noticeable with polCA. The curve for polEP is not suitable for 50% of the samples, unaffected by storage. A regression curve could be found for all samples in T01 and T02 for acrEP and acrCA. Only when applying the tare from T01 to T02 in T02_T three curves in acrEP and one curve in acrCA fall outside the regression criterion. It is worth noting that all samples of the BL storage slip from the polynomial regression into the linear regression in acrCA. This could be attributed to a weaker bond between the steel and fibre. The same is true for specimens polCA. As for acrEP, storing it in T02 does not appear to have any significant impact on its regression classification.

The width of the corridor between the highest and lowest measured value for each *measurement range* is the first quantitative criterion analysed. 0–100 (100), 100–200 (200), 200–300 (300), and greater than 300 μm/m (+). Although this categorisation is arbitrary, the smallest possible range is preferred. It is noticeable that for acrCA, the strain corridor is smaller than 200 μm/m across all values. The value does not increase much with storage. There is a slight degradation in the value for acrEP. The corridor before and after storage is comparatively higher for polEP than for the acrylate-coated fibres. Storage has a slight influence on this, but it is not reproducible in one direction. For polCA, storage has a negative effect on the strain range for all samples, e.g., over 400 μm/m when stored in BL.

With the *mean strain difference* the mean values of the strain over time and length between T01 and T02 are compared. Values less than 0 indicate that T02 is greater than T01 and vice versa. A good result is achieved if the tests T01 and T02 are repeatable. Therefore, the differences are categorised into four limits, with the narrowest limit always being the most significant. The strain values range from −20 to +20 (±20), −40 to +40 (±40), and −60 to +60 (±60) μm/m, with outliers beyond this range (±). PolEP produces the best result, with only one value outside the ±40 range. AcrEP, however, has two outliers when stored in WL, making this combination the worst. AcrCA and polCA both exhibit one outlier, yet the former stands out with a consistently mean value of strain. Compared to the mean value of T01 shown in Figure 17b, these maximum variances are relatively small (LLacrCA02: $\frac{78\;\mu m/m}{1300\;\mu m/m}=0.06$. It is possible that this error could originate from a clamping error and an underlying residual stress.

Another approach aims to assess the time-dependent stability of the curves. The relative *standard deviations of the regression curves* at each time value provide a reliable measure for determining temporal stability. Purely considering the mean values and their standard deviation spatially and temporally would not account for the shape of the strain curve. The results in Table 3 are categorised as <0.1% (0.1), 0.1–1% (1), 1–10% (10), and greater than 10% (+). The strain of the acrCA samples appears to be unaffected by storage, similar to acrEP. However, the worst value for acrEP scatters more widely than for acrCA. LLpolEP01T02 shows poor quality measurement data (Figure 13), causing the regression to jump and become dissimilar. Overall, the combination is average. PolCA produces the worst result, with any storage affecting the constancy of the measured values. One has to note that a logarithmic scale is used. The relative standard deviations vary within a factor range of $10^{4}$.

The final criterion used is the *$R^{2}$ vector distance* from T01 to T02 or T02_T. The smaller the Euclidean distance, the better the match between the strain curves’ characteristics. The table’s gradations are categorised as follows: less than 0.33 (0.33), 0.33–0.66 (0.66), 0.66–1 (1), and greater than 1 (+). When using the polEP combination, at least one equivalent regression curve (T02 or T02_T) can be formed for all measurement curves T01. These measurements are generally stable even after storage, except for WLpolEP02T01 where a good regression curve cannot be found. However, T02 has a good regression curve, which is why the distance is high. The delta is lower and therefore better for T02_T. It is worth noting that the choice of tare has a significant impact on the measurements of polAcr. LLpolCA01 scatters strongly after storage with either tare. BLpolCA03 consistently shows poor regression (all $R^{2}$ values are low), resulting in a small distance. It is not possible to determine which tare is better overall. LLpolCA01 exhibits poor regression on T02_T, but performs better on T02 than on T01. For SLpolCA03, T01 performs well initially, but shows poor performance after storage. Storage has no significant impact on WL. Most of the acrEP specimens have regression curves that are equally good or bad, except for the value of WLacrEP03T02_T which stands out. The measured values do not match the tare of T01, which may be due to residual stresses from the first test.

It is noticeable that the regression curves change from 3rd degree to 2nd degree due to the storage in the liquids (SLacrEP01, WLacrEP02) or a tare that does not fit (SLacrEP02). The regression curves of LLacrEP02, SLacrEP03 and WLacrEP01 fit each other well. The vector distances in acrCA are comparatively large due to the dominant regression curve often changing from polynomial to linear (BLacrCA, WLacrCA03). Moisture could impair the bond, resulting in imprecise strain transfer or equalisation. The shape of the curves remains constant in condition LL, except for LLacrCA03, where only the position changes slightly. The same applies to SL. There is only one outlier at SLacrCA01T02_T, which could again be caused by an error in the tare (residual stress). The scores result from the vector distances, without the influence of interpretative explanations.

## 6. Discussion of the Previous Experiments

The tests on duct presented in Section 4 and the environmental exposure tests in Section 5 have demonstrated, that the use of DFOS in concrete can be sophisticated. On one hand, this is demonstrated by the application of fibres to host materials. When it comes to ducts, winding the fibres requires effort. While cyanoacrylate can be used as an adhesive, it needs for contact pressure that is difficult to apply during winding. Although 2k epoxy could be a potential alternative, it was not within the scope of this particular study. The measurement data is of poor quality due to the residual stress introduced during the step-by-step bonding process of the fibre attachment. The bond to the carrier material cannot be guaranteed throughout. Additionally, the fibre used contained many SRAs. It may be more suitable to use nylon or acrylate fibres for wrapping. Furthermore, the fibre must be protected on the duct to withstand the harsh conditions during concreting. The efficacy of silicone as a sealant requires further investigation. The investigated method does not appear to be a practical solution for ducts in structures, as it is a complex manufacturing process, has a limited monitoring range in comparison to the sensor length, can potentially lead to SRAs due to the tight windings, the results are difficult to reproduce, and the reliability of the measured data is questionable. Better ways to measure strains on tendons include the application of sensor cables near the tendon strands or the installation of fibre optic sensors in the strands, as is being pursued in the smart tendon research project [23], for example, or is already being used with SmART Strands [21].

Attaching the fibres to the sensor rods is a simple process, as each rod can be manufactured in one piece. This ensures that the fibre only experiences the strain applied for tensioning and the strain from shrinkage introduced by the adhesive during hardening. Although there may be slight differences in strain due to variations in adhesive thickness and wetted surface, these can be eliminated by tare. Errors may occur when misusing the 2k epoxy adhesive. Small quantities can result in different compositions of the two components, leading to varying material properties of the adhesive. As noted in [46], one component may remain unmixed even when sprayed. If the sensor rods are used in structures, the bonding must be protected with a silicone sealant, for example, as in laboratory tests. In more challenging construction environments, mechanical influences should not be overlooked, as discussed in Section 5.1. Initial tests were conducted on a bridge for the sensor rods [35,36]. The chemical influences proved to be less problematic than the harsh environment, which resulted in damage to the pigtails and the sealed rods. Cables that can be simply led out of the structure may offer a viable alternative.

The production of grooves in advance is a significant issue and root of uncertainties. Manual production does not allow for good precision, resulting in varying cutting depth and alignment in the longitudinal direction due to the scattering of hardness resulting from the rolling process of the reinforcement. While a rail supports for short bars, the wear on the cutting device is too high for longer bars, necessitating readjustment of the depth. Alternatively, methods using a planer are suggested and executed in [7,50]. The wide variation in slot design may be one reason for the differing results. A groove measuring approximately 1–2 mm in width and 1–4 mm in depth indicates an absolute cross-sectional loss of 1–8 ${\mathrm{mm}}^{2}$. This corresponds to a cross-sectional loss of 0.5–7.1%. As a result, strains may increase within a certain range under a constant load. The suitability of the criterion of regression curves could be debated here. The scattering range, which appears wavy in Figure 14a,b, may also be due to uneven grooves. This has a significant impact on the strain difference between the maximum and minimum values over time, as shown in Figure 16. Upon further examination of the mean value curve, it is evident that its range does not deviate greatly from the absolute range, even when suppressing temporal variability. High strain differences between the two ranges in Figure 21 are only occasionally visible. These differences are caused by temporal drift.

Both investigations were conducted using the same measuring device. Due to the duct’s thin-walled sheet metal profile, it was not feasible to clamp it directly in the test rig for a tensile test. Therefore, a device (Figure 6a) was designed to apply the load to the duct. However, the loads were limited as shown in Figure 6b due to high plastic deformations. Maintaining the load was only possible through a large applied path in the testing machine. The distribution of plastic deformations between the duct geometry and load-bearing points could not be determined. If further investigations are required, a different test setup would be necessary.

A clamping support designed for round steel tensile specimens was used. This support rigidly fixes the specimens and ensures they are perfectly aligned. Reinforcing bars may have a manufacturing tolerance due to rolling, which can cause curvature. During fixation a new preload is imposed on them as the clamping jaws straighten the possibly crooked specimen ends. However, when tension is applied, the specimen deflects. This can lead to linear or polynomial deformation and strain curves for both the tare and the load, as all internal forces can be absorbed via the rigid bearings. The test results for T02 and T02_T show significant deviation when the specimen is installed with slight rotation, but only shifted curves for T01 when installed identically. Although the front side of the samples was marked during installation, some of the markings were lost during storage. Therefore, it was not possible to guarantee completely identical specimen installation. If the test series were to be repeated, this optimisation should be considered.

Previous findings have shown that the uncertainties in manufacturing the sensor rods are so high that they are not well suited for precise strain measurement in manual production. Furthermore, a hinged installation of the specimens in the testing machine would be better.

It is evident that protecting the fibre/adhesive combination from external influences is crucial, especially in the case of ducts that are subjected to high mechanical loads. In the tests on the sensor rods, unsealed configurations were used; mechanical protection is provided by the groove. The tests conducted were chemical in nature. When bonding with cyanoacrylate, the fibre is not fully embedded in the adhesive, allowing contact with the surrounding medium. This is particularly evident in the case of polyimide fibres, as described in [10], where they are susceptible to attack by alkaline substances. Additionally, refs. [51,52,53] demonstrate that polyimide fibres interact with moisture. The presented results also indicate that polyimide fibres require adequate protection, such as an epoxy bond, to function effectively. The polyimide-coated and acrylate-coated fibres each showed only one failed sensor, indicating their reliability. Epoxy bonding also provides mechanical protection. Acrylate fibres have a stable coating and can function effectively without a protective layer. When combined with cyanoacrylate, they consistently produce good measured values, as summarised in Table 3. It is noteworthy that the regression curves for cyanoacrylate combinations often shift from polynomial to linear. For this reason, the adhesive structure may have been altered due to the liquids used in the storage tests.

Finally, it is important to name the evaluation methods used. Currently, the authors are not aware of any freely accessible Python frameworks for the general data evaluation of the fibre optic measurement data from this measurement system. The pre-processing and categorisation methods used in this research were developed as part of the study. They are tools that use known statistical methods to compare data of the same origin. We welcome any additional ideas for analysing the investigations and are willing to provide the data.

## 7. Conclusions and Outlook

We present the reasons and current efforts to incorporate Distributed Fibre Optic Sensors (DFOS) into concrete. Researchers are studying the bonding behaviour of fibres on concrete, as well as the reinforcing bars. Robust sensor solutions have been proposed, but results are typically limited to laboratory settings. The fibres used are usually coated with polyimide, acrylate, polymers, or nylon. The bond with the host material is established using cyanoacrylate, 2k epoxy or polymer-based adhesives. Host materials included concrete, reinforcing steel, plastics or tendons. Two approaches were investigated to measure the strains in concrete. In the first experiment, a polyimide-coated fibre was bonded to a tendon duct with cyanoacrylate to measure the strains on the host material duct and monitor the strain transfer from the tendon to the concrete. The method was found to be overly complex, lacking robustness, and producing inaccurate measurements. Therefore, it is not considered suitable in its current form. In a subsequent test, the reinforcement bars were bonded with fibres that have already been investigated by other research groups. In addition to already investigated mechanical influences, we analyse the chemical effects of environmental factors that occur during the life cycle of a reinforcing bar. These factors include air storage, water storage, saline storage, and exposure to an alkaline environment. The test specimens consist of a reinforcing bar with a groove, to which a fibre (with polyimide or acrylate coating) is bonded using an adhesive (cyanoacrylate or 2k epoxy). This study involved the use of three sensor rods per one out of four sensor configurations, with each specimen being stored in one of four different environments after an initial tensile test (T01). Following storage, the 48 specimens were subjected to a second tensile test, and the resulting strain curves were compared both qualitatively and quantitatively using optical comparison, statistical mean value calculations, and applied regression analysis. The study demonstrated that the acrylate fibres bonded with cyanoacrylate exhibited the highest resistance to chemical effects, while the polyimide-coated fibres bonded with cyanoacrylate showed the lowest. However, the manufacturing process of the sensor rods is subject to many unknowns, such as high tolerances in the production of the reinforcing steel, the application of the groove to the rods, and the bonding procedure. As a result, measurement errors can accumulate quickly.

The research team has decided not to pursue external application for ducts due to its limited effectiveness. Instead, further experiments will be carried out with solutions for strands. The experiments with the sensor rods have shown that while there are good combinations, the rods are subject to many mechanical and chemical influences. Sensor cables can help avoid problems. However, the production, installation, and robustness of the sensor rods raise major questions. Even on a laboratory scale, the rods cannot be produced reliably. Additionally, research is being conducted on the integration of fibre optic big data for structural health monitoring and non-destructive testing. The emphasis in the former is on automatic integration into digital twins.

## Figures and Tables

**Figure 1 sensors-24-06122-f001:**
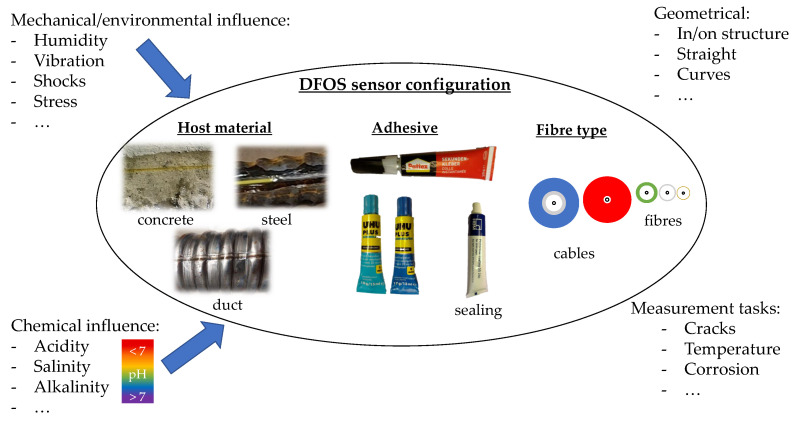
Components and influences on a DFOS sensor configuration.

**Figure 2 sensors-24-06122-f002:**
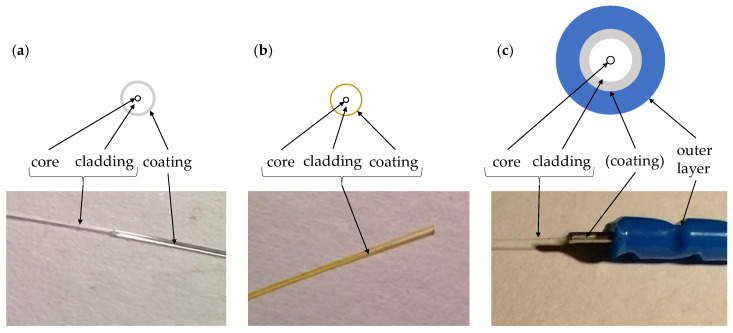
Fibre types, sketches and pictures. (**a**) Acrylate coated fibre. (**b**) Polyimide coated fibre. (**c**) Polyamide fibre optic cable [6].

**Figure 3 sensors-24-06122-f003:**
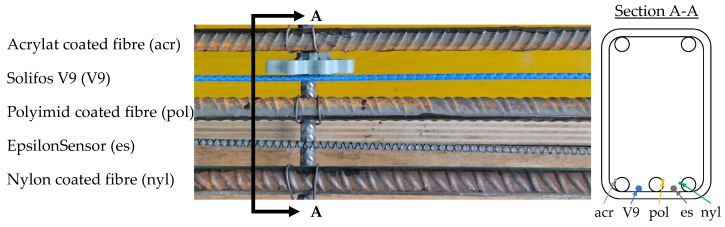
Different cable and fibre types in a reinforcement cage.

**Figure 4 sensors-24-06122-f004:**
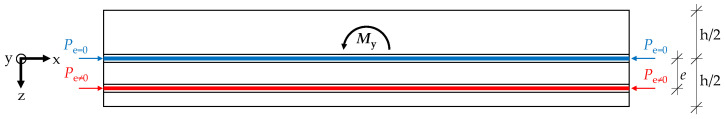
Prestressed concrete beam, principle sketch of centric (blue, $P_{e=0}$) and excentric (red, $P_{e\;\neq \;0}$).

**Figure 5 sensors-24-06122-f005:**
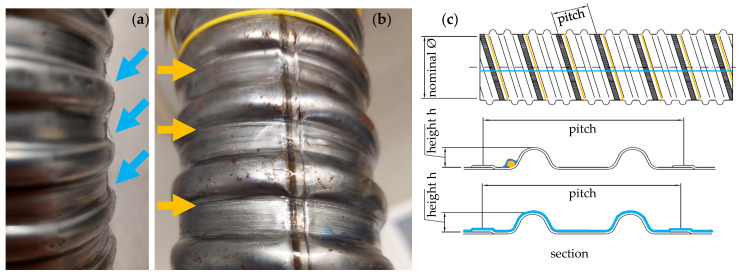
Methods of fibre attachment onto ducts: (**a**) Longitudinal direction (fibre marked by blue arrows), (**b**) Along the coil (marked by yellow arrows), (**c**) Sketches in view and longitudinal section with blue longitudinal fibre and yellow coil fibre, drawing by [38].

**Figure 6 sensors-24-06122-f006:**
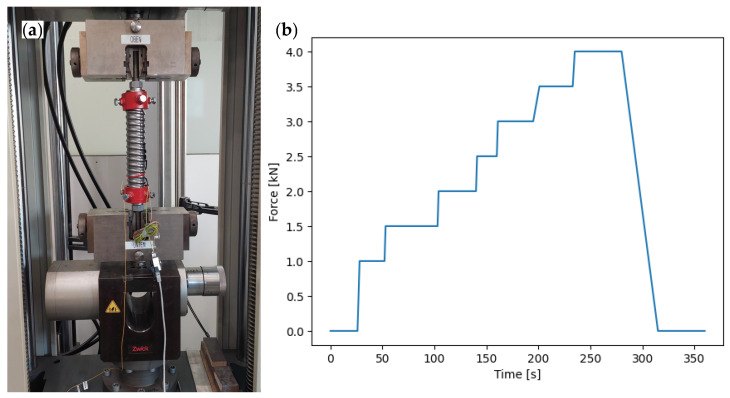
Experimental setup: (**a**) Fixed specimen, (**b**) Load curve of test.

**Figure 7 sensors-24-06122-f007:**
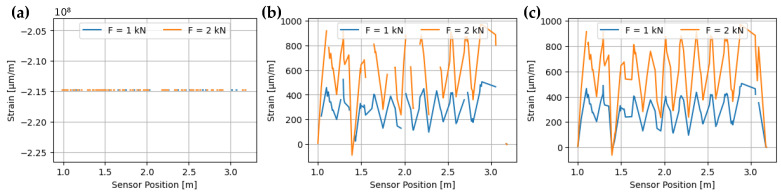
Steps of data processing (**a**) Raw data, (**b**) Cleaned data, (**c**) Smoothed data.

**Figure 8 sensors-24-06122-f008:**
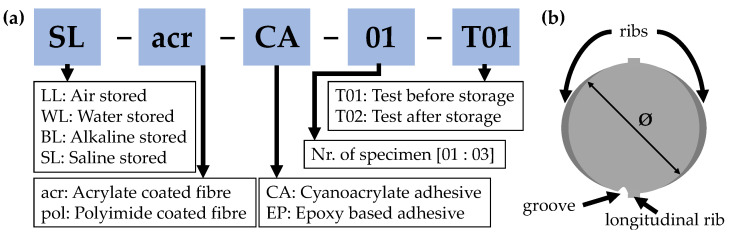
(**a**) Overview of the specimens configurations, (**b**) Structure of the prepared rebar.

**Figure 9 sensors-24-06122-f009:**
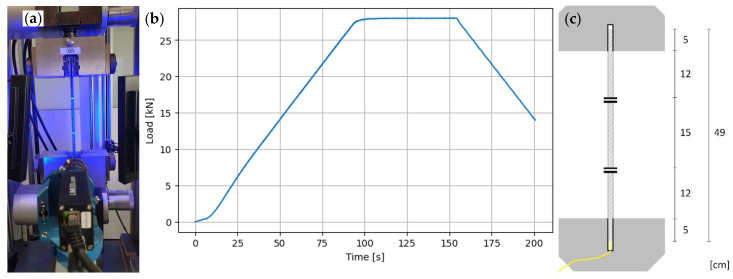
Experimental setup of rebar tensile tests (**a**) Fixed specimen, (**b**) Load curve of test, (**c**) Specimen in Zwick Z400 with measurement marks for RTSS.

**Figure 10 sensors-24-06122-f010:**
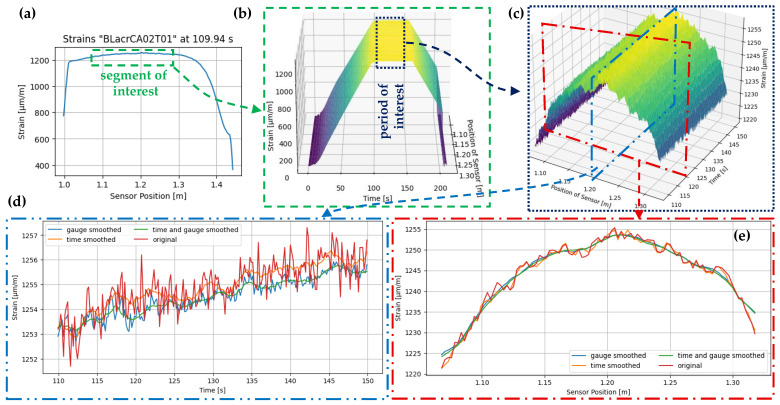
Steps of data processings (**a**) Raw data, (**b**) Cut out segment of interest, (**c**) Cut out period of interest, (**d**) Strain over time for one single gauge (raw and smoothed data), (**e**) Strain at the length of sensor at a single timestamp (raw and smoothed data).

**Figure 11 sensors-24-06122-f011:**
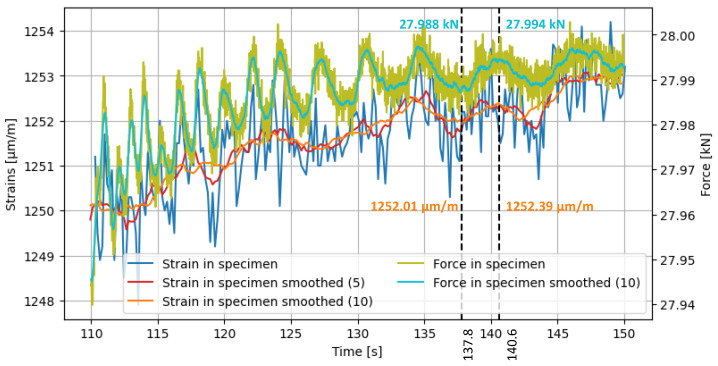
Comparison of strain and smoothed strain over time at a specific gauge and force and smoothed force over time in a specimen.

**Figure 12 sensors-24-06122-f012:**
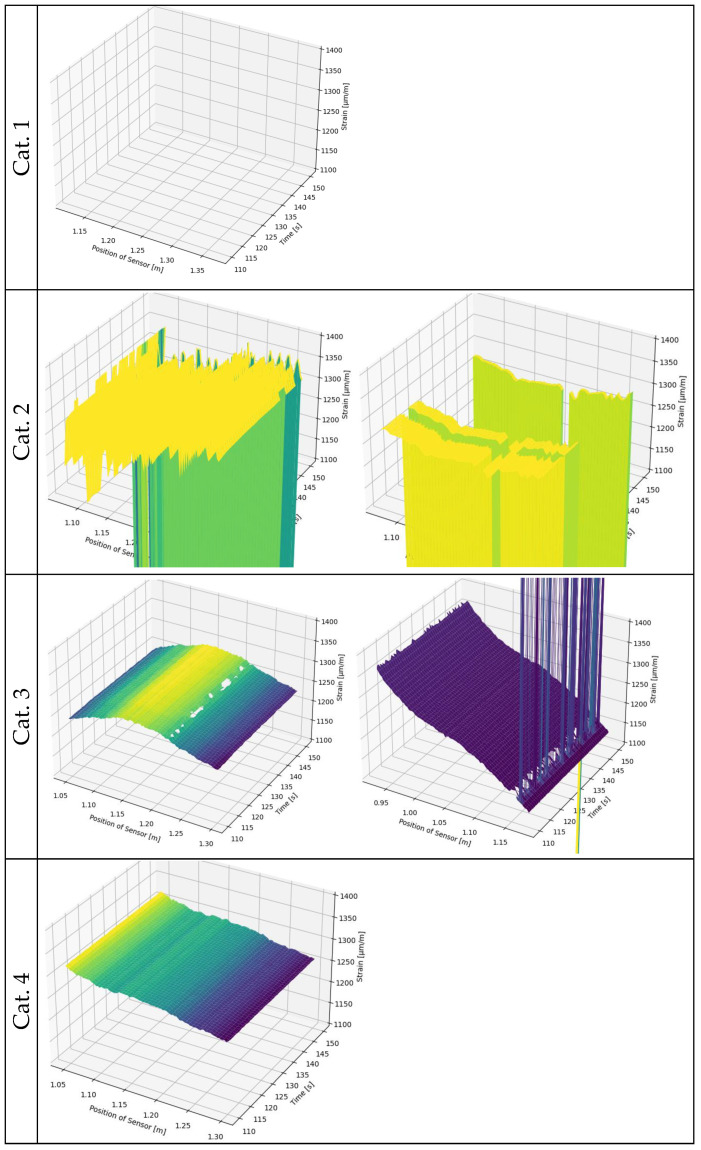
Categories of measurement results.

**Figure 13 sensors-24-06122-f013:**
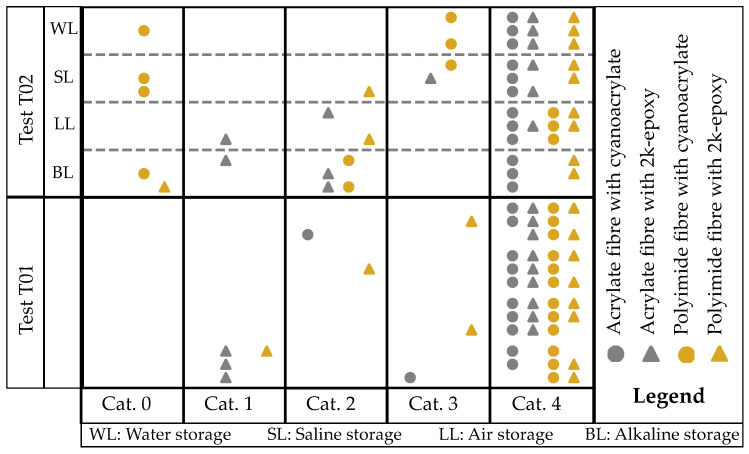
Qualitative results of the tensile tests T01 and T02, arranged by category and sensor configuration.

**Figure 14 sensors-24-06122-f014:**
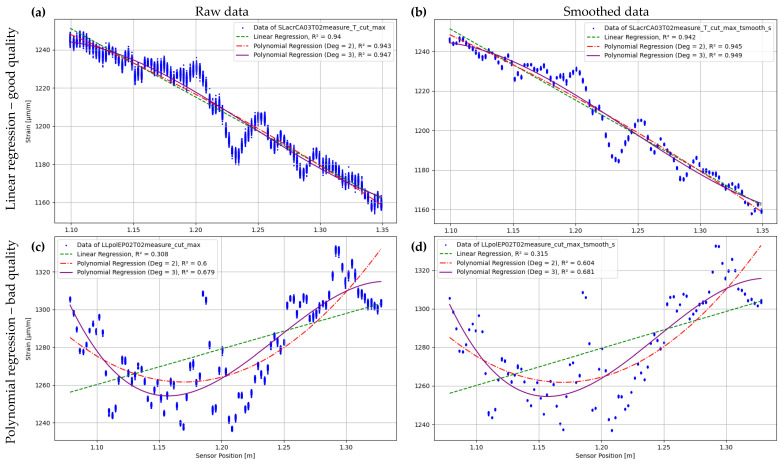
Classification of measurement curves of raw data and smoothed data (**a**) linear regression with raw data, (**b**) linear regression with smoothed data, (**c**) polynomial regression with raw data, (**d**) polynomial regression with smoothed data.

**Figure 15 sensors-24-06122-f015:**
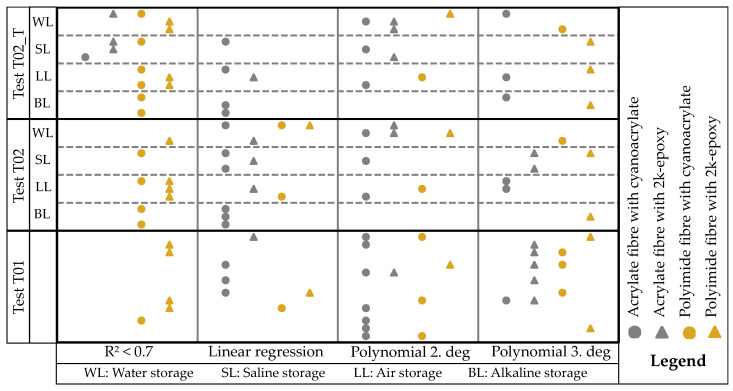
Classification of the smoothed measurement curves into the best fitting regression curve for the three test series.

**Figure 16 sensors-24-06122-f016:**
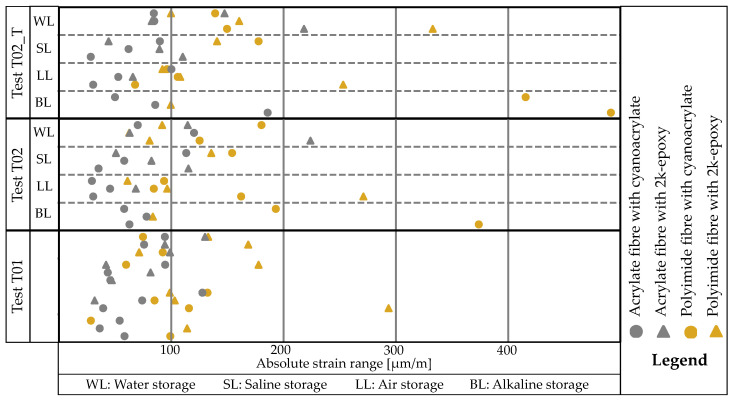
Strain ranges of the specimens for the three test series.

**Figure 17 sensors-24-06122-f017:**
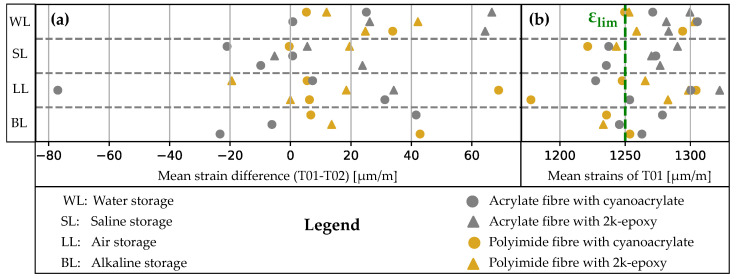
(**a**): Mean strain differences ΔØ(T01−T02) for smoothed data, (**b**): Mean strains of T01 measurements.

**Figure 18 sensors-24-06122-f018:**
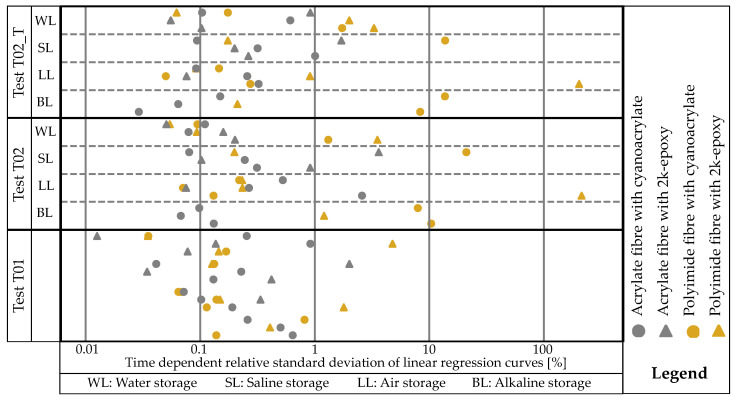
Relative standard deviations over time of the curves from linear regression.

**Figure 19 sensors-24-06122-f019:**
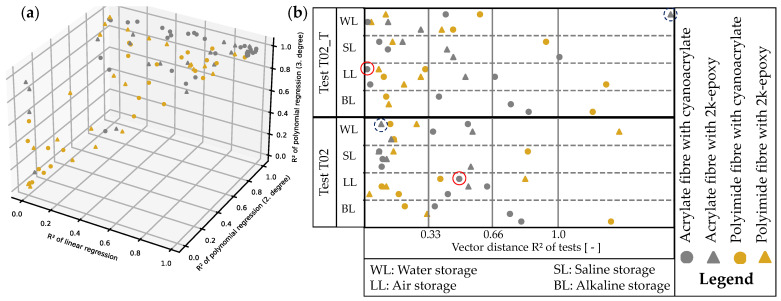
(**a**) Visualisation of the $R^{2}$ vectors in space, (**b**) Euclidean distances of the $R^{2}$ vectors to T01 with the marked WLacrEP03 (grey circle, dotted) and LLacrCA03 (red circle, full).

**Figure 20 sensors-24-06122-f020:**
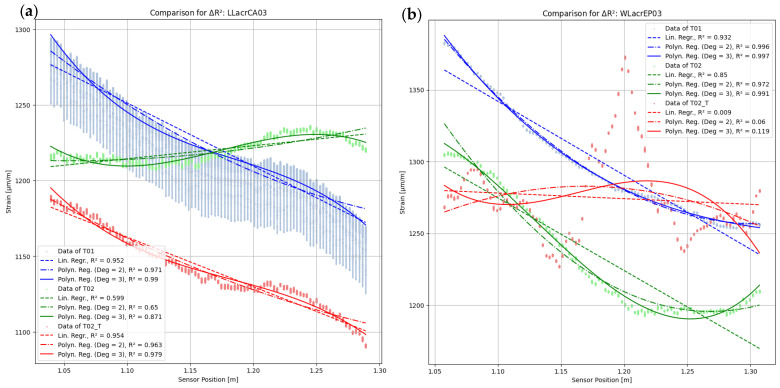
Exemplary regression curves of two specimens (**a**) LLacrCA03, (**b**) WLacrEP03.

**Figure 21 sensors-24-06122-f021:**
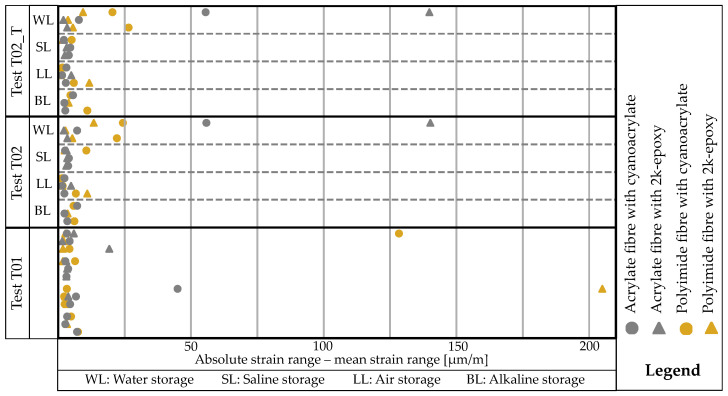
Difference of absolute strain range—mean strain range of specimen for the three test series.

**Table 1 sensors-24-06122-t001:** Performance differences between the various distributed sensing techniques (from [4]).

Sensing Technology	Transducer Type	Sensing Range	Spatial Resolution	Main Measurands
Raman OTDR	distributed	1–37 km	1 cm–17 m	temperature
Brillouin OTDR	distributed	20–50 km	≈1 m	temperature, Strain
Brillouin OTDA	distributed	150–200 km	2 cm (2 km)–2 m (150 km)	temperature, strain
Rayleigh OFDR/DAS	distributed	50–70 m	≈1 mm	temperature, strain
FBG	quasi-distributed	≈100 channels	2 mm (Bragg length)	temperature, strain, displacement

**Table 2 sensors-24-06122-t002:** Factors $k_{\mathrm{pitch}}$ for SUSPA DSI ducts with data from [38].

Parameter	Duct Attributes						
$\phi_{\mathrm{nominal}}$ [mm]	40	45	…	105	110	115	120
$l_{\mathrm{pitch}}$ [mm]	26	26	…	26	24	24	24
$k_{\mathrm{pitch}}$	0.171	0.155	…	0.073	0.065	0.062	0.060

**Table 3 sensors-24-06122-t003:** Scores per sensor configuration summarised from the evaluation of the qualitative and quantitative measurement results with final score.

	Fibre		Polyimid		Acrylate	
	Bonding		2k Epoxy	Cyanoacrylate	2k Epoxy	Cyanoacrylate
Figure 13	Category	T02	1, 0, 2, 0, 9	4, 0, 2, 3, 3	0, 2, 3, 1, 6	0, 0, 0, 0, 12
	0, 1, 2, 3, 4	T01	0, 1, 1, 2, 8	0, 0, 0, 0, 12	0, 3, 0, 0, 9	0, 0, 1, 1, 10
	Score category		3	1	2	4
Figure 15	Regression type	T02_T	4, 0, 1, 3	6, 0, 1, 1	3, 1, 3, 0	1, 4, 3, 3
	0, lin, pol2, pol3 *	T02	4, 1, 1, 2	4, 2, 1, 1	0, 3, 2, 2	0, 6, 3, 2
		T01	4, 1, 1, 2	1, 1, 3, 3	0, 1, 1, 5	0, 3, 7, 1
	Score regression type		2	1	3	4
Figure 16	Range of	T02_T	3, 3, 1, 1	2, 4, 0, 2	4, 2, 1, 0	10, 1, 0, 0
	measurement	T02	6, 1, 1, 0	2, 5, 0, 1	4, 2, 1, 0	9, 2, 0, 0
	100, 200, 300, + **	T01	2, 5, 1, 0	6, 2, 0, 0	6, 1, 0, 0	10, 1, 0, 0
	Score measurement range		2	1	3	4
Figure 17	Mean strain difference T01−T02		6, 1, 1, 0	5, 1, 1, 1	2, 3, 0, 2	5, 4, 1, 1
	±20, ±40, ±60, ± ***					
	Score mean strain difference		4	2	1	3
Figure 18	Standard deviation	T02_T	2, 3, 2, 1	1, 3, 2, 2	2, 4, 1, 0	4, 6, 1, 0
	of linear regression	T02	2, 3, 2, 1	2, 2, 2, 2	2, 4, 1, 0	4, 6, 1, 0
	0.1, 1, 10, +% ****	T01	2, 4, 2, 0	2, 6, 0, 0	3, 3, 1, 0	2, 9, 0, 0
	Score standard deviation		2	1	3	4
Figure 19	$R^{2}$ Vector distance	T02	7, 1, 0, 0	3, 2, 1, 2	3, 3, 0, 1	5, 2, 3, 1
	0.33, 0.66, 1, + *****	T01	6, 0, 1, 1	5, 1, 1, 1	4, 3, 0, 0	3, 6, 2, 0
	Score $R^{2}$-Distance		3	1	4	2
	**Final Score** ******		**16**	**7**	**16**	**21**

* Categories by Figure 15: 0: no category, lin: linear regression, pol2: polynomian regression 2. degree, pol3: polynomial regression 3. degree. ** Ranges up to 100, 200, 300 or more (+) μm/m. *** Differences of the mean strain of measurement Δ=T01−T02 up to ±20, ±40, ±60 or more (±) μm/m. **** Relative standard deviations of linear regression curve parameters, less than 0.1, 1, 10 or higher than 10 %. ***** Euclidic distances of the $R^{2}$-Vectors of the regression curves, less than 0.33, 0.66, 1 or higher than 1. ****** Sum of all the score rows.

## Data Availability

The raw data supporting the conclusions of this article will be made available by the authors on request.

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
