# Peer review of "Investigation of the Robust Integration of Distributed Fibre Optic Sensors in Structural Concrete Components"

_sensors, 2024, doi:10.3390/s24186122_

Round 1

Reviewer 1 Report

Comments and Suggestions for Authors

This paper provides valuable insights into the integration of distributed fiber optic sensors in concrete structures. However, it has several areas needing further clarification and improvement:

1. Suggest including all measurable variables in Table 1 for clarity and completeness.

2. The literature review is comprehensive but does not sufficiently highlight the unique contributions of this paper. Needs a more explicit connection between past research and the current study.

3. The author mentioned NaN and unrealistic strain peaks occurring in Figure 7. Did the author conduct additional installations to confirm the guessed reasons since only one test was done?

4. The paper notes inconsistent curing conditions for adhesives, which affect bonding. However, it does not propose a controlled methodology to ensure consistent adhesive application.

5.The paper indicates that softer adhesives bridge cracks and protect fibers but compromise strain sensitivity. Is there an optimized balance between adhesive stiffness and strain sensitivity?

6.The paper frequently mentions SRAs without delving deeply into their root causes or providing a robust solution.

7.The real-world applicability under varying construction and operational conditions needs to be further discussed.

Author Response

Comments 1: Suggest including all measurable variables in Table 1 for clarity and completeness.
Answer 1:
We added the column as recommended in Table 1.

Comments 2: The literature review is comprehensive but does not sufficiently highlight the unique contributions of this paper. Needs a more explicit connection between past research and the current study.
Answer 2: Agree, we added "The literature study showed that ducts have not yet been used as host materials for DFOS. Furthermore, there is a lack of chemical-based studies on the use of DFOS on rebars." in chapter 2.4 – last paragraph (L164f).

Comments 3: The author mentioned NaN and unrealistic strain peaks occurring in Figure 7. Did the author conduct additional installations to confirm the guessed reasons since only one test was done?
Answer 3: Like already mentioned in Chapter 7 we will not “pursue external application for ducts due to its limited effectiveness” in future research. The tensile test was the only one performed. No other tests or installations were planned due to the reasons listed before: “The method was found to be overly complex, lacking robustness, and producing inaccurate measurements”. For clarification purposes, we have added the following to the end of section 4.4: “No additional sensors were installed as this was a feasibility study.” (L321f)

Comments 4: The paper notes inconsistent curing conditions for adhesives, which affect bonding. However, it does not propose a controlled methodology to ensure consistent adhesive application.
Answer 4: You are correct, no methodology is proposed. However, the paper never claimed to provide a best practice guide for the best application for bond. We explained our approach and the results out of it.

Comments 5: The paper indicates that softer adhesives bridge cracks and protect fibers but compromise strain sensitivity. Is there an optimized balance between adhesive stiffness and strain sensitivity?
Answer 5: The sentence “The ideal stiffness of the bond is a compromise between sensitivity and strain.” is taken from chapter 2.3. The optimum combination could not be found in the literature study. However, crack monitoring is not within the scope of the paper.

Comments 6: The paper frequently mentions SRAs without delving deeply into their root causes or providing a robust solution.
Answer 6: Other researchers like [31] studied this topic in more detail and focus on this issue. In our research, like shown for instance in Figure 12, an SRA is only an indicator for the quality of a sensor, without knowing its root.

Comments 7: The real-world applicability under varying construction and operational conditions needs to be further discussed.
Answer 7: Agree. We added some sentences in the discussion chapter, where we also targeted this topic.
Paragraph 1: “The investigated method does not appear to be a practical solution for ducts in structures, as it is a complex manufacturing process, has a limited monitoring range in comparison to the sensor length, can potentially lead to SRAs due to the tight windings, the results are difficult to reproduce, and the reliability of the measured data is questionable.” (L818ff)
Paragraph 2: “If the sensor rods are used in structures, the bonding must be protected with a silicone sealant, for example,  as in laboratory tests. In more challenging construction environments, mechanical influences should not be overlooked, as discussed in chapter 5.1. Initial tests were conducted on a bridge for the sensor rods [35,36]. The chemical influences proved to be less problematic than the harsh environment, which resulted in damage to the pigtails and the sealed rods. Cables that can be simply led out of the structure may offer a viable alternative.” (L833ff)

Reviewer 2 Report

Comments and Suggestions for Authors

This manuscript presented the reasons and current efforts to incorporate distributed fiber optic sensors (DFOS) into concrete. The authors studied the bonding behavior of fibers on concrete, as well as the reinforcing bars. Meanwhile, two approaches were provided to detect the trains in concrete. The experimental procedure is detailed, including considerations for embedding fibers in concrete structures, the approach to embedding tendons with hollow tubes, the experimental setup, and the results of experiments on rebars, among others. It has a significant reference value for the related devices in practical applications. Overall, I recommend this work be published when the suggestions listed below are adequately addressed.

 1. There are issues with the capitalization of some proper nouns. For example, "Civil Engineering Structural Health Monitoring" should be "civil engineering structural health monitoring," and "Raman Optical Time Domain Reflectometry" should be "Raman optical time domain reflectometry" et al.

2. How can we prevent the aging and performance degradation of devices in practical applications? Also, how do we maintain the authenticity of data as devices degrade?

3. How to prevent cross-sensitivity problems in the detection process, such as strain measurement, and eliminate the impact of temperature and humidity on the data should be explained in detail.

Comments on the Quality of English Language

There are issues with the capitalization of some proper nouns.

Author Response

Comments 1: There are issues with the capitalization of some proper nouns. For example, "Civil Engineering Structural Health Monitoring" should be "civil engineering structural health monitoring," and "Raman Optical Time Domain Reflectometry" should be "Raman optical time domain reflectometry" et al.
Answer 1: Agree, we have written the capital letters in lower case.

Comments 2: How can we prevent the aging and performance degradation of devices in practical applications? Also, how do we maintain the authenticity of data as devices degrade?
Answer 2: We understand your point. Discussing this is not within the scope of this paper. However, we have included it in Chapter 2.5, last paragraph at the end: “The consideration of measurement errors (bond, ageing of sensor and measuring device) should not be neglected [34], but is beyond the scope of this paper.” (L179f)

Comments 3: How to prevent cross-sensitivity problems in the detection process, such as strain measurement, and eliminate the impact of temperature and humidity on the data should be explained in detail.
Answer 3: Agree. Temperature and humidity have an influence on the fibre and therefore the measurement. However, the tests presented in this paper deal with the evaluation of short-term tests under constant conditions in the laboratory. Temperature and humidity influences do not play a role here. We will therefore not go into this topic. In addition, it is already addressed in the literature study.

Reviewer 3 Report

Comments and Suggestions for Authors

The paper is a comprehensive and extended study of the effectiveness of integrating Rayleigh-based DFOS on tendons in concrete structures.

While the manuscript has merit, it is clear that it needs a major revision to be accepted. This revision is crucial to ensure the manuscript meets the required standards and is ready for publication. 

The following issues need for attention:

1) The Authors are recommended to reduce the section/s related to the radial integration: the results show this solution is not viable. I understand that this aspect needs to be motivated, but the reader is led to initially believe that this configuration is the one that is chosen. The unfeasibility could be forecasted in advance as:

a) the deployment requires a lot of effort

b) the length of the bar, which is monitored is significantly reduced with respect to the fiber deployed

c) One has to deal with bending-induced losses (these could explain the large number of SRAs)

d) the deployment suffers from reproducibility and inaccuracies due to the bar surface inhomogeneity

e) the raw data collected is very "far" from the target one (bar longitudinal elongation/strain/stress), and requires numerical manipulation, which inevitably introduces numerical noise and errors 

Moreover, this radial solution would have been novel, while the final one, with the fiber within a groove, is the classical one already explored. This greatly impairs the novelty of the manuscript.

2) I cannot see the point in averaging the raw data, at least for the averaged strain levels addressed in this study; e.g., in Fig 10 (d), (e), the strain fluctuates by a few ue over more than 1000 ue, which is less than 0.5%. Do the authors really need to remove such fluctuation? Additionally, couldn't the time fluctuation be due to the tensile test apparatus?

3) I cannot understand how a tensile test can produce a non-flat strain. If one pulls a bar from its ends, the strain is almost constant across the bar.

E.g., Fig 12, cat.3  left: in this case, the strain is parabolic, which means that the bar is bending; Fig. 15 (a): in this case, the strain varies linearly, with a variation across the bar is more than 10%, which is well above the acceptable one). This may be due to a slippage in one of the clamps. Again, in Fig. 15 (c), the strain fluctuation may be due to a local wrong alignment of the fiber. 

In both cases,  it means that the authors do not fully control the setup.

An additional minor issue is the following: subplots should be always labeled following a clockwise sorting.

Author Response

Comments 1: The Authors are recommended to reduce the section/s related to the radial integration: the results show this solution is not viable. I understand that this aspect needs to be motivated, but the reader is led to initially believe that this configuration is the one that is chosen. The unfeasibility could be forecasted in advance as:

  1. the deployment requires a lot of effort
  2. the length of the bar, which is monitored is significantly reduced with respect to the fiber deployed
  3. One has to deal with bending-induced losses (these could explain the large number of SRAs)
  4. the deployment suffers from reproducibility and inaccuracies due to the bar surface inhomogeneity
  5. the raw data collected is very "far" from the target one (bar longitudinal elongation/strain/stress), and requires numerical manipulation, which inevitably introduces numerical noise and errors

Moreover, this radial solution would have been novel, while the final one, with the fiber within a groove, is the classical one already explored. This greatly impairs the novelty of the manuscript.
Answer 1: Agree, we have incorporated these aspects into the discussion, or rather emphasised them better. Chapter 6, first paragraph: “The investigated method does not appear to be a practical solution for ducts in structures, as it is a complex manufacturing process, has a limited monitoring range in comparison to the sensor length, can potentially lead to SRAs due to the tight windings, the results are difficult to reproduce, and the reliability of the measured data is questionable.” (L818f)
We have also incorporated points b) and c) in Chapter 4.2. Further points are difficult to predict, as they only become apparent during the execution and evaluation of the experiment. For this reason, we have not forecasted these points. “For this reason, this variant was not considered further, and only variant number two (Figure4b) was pursued although it is not ideal either. A long sensor length is required to cover a comparatively small section of the tendon. In addition, the tight bending radii are a possible source of SRAs.” (L261ff).

Comments 2: I cannot see the point in averaging the raw data, at least for the averaged strain levels addressed in this study; e.g., in Fig 10 (d), (e), the strain fluctuates by a few ue over more than 1000 ue, which is less than 0.5%. Do the authors really need to remove such fluctuation? Additionally, couldn't the time fluctuation be due to the tensile test apparatus?
Answer 2: Agree. The fluctuations in this curve (Figure 10d) are small. However, in other curves, which are classified in other categories (Table 3), they vary further. We apply the same smoothening to all curves. In the example of Figure 10, a data set with good data quality was used to avoid confusion. Figure 14 shows larger deviations in the percentage range, but these are not yet the smallest. Figure 11 shows that these temporal fluctuations are induced by the force. This has already been described there. In our view, no revision is necessary.

Comments 3: 3) I cannot understand how a tensile test can produce a non-flat strain. If one pulls a bar from its ends, the strain is almost constant across the bar.

E.g., Fig 12, cat.3  left: in this case, the strain is parabolic, which means that the bar is bending; Fig. 15 (a): in this case, the strain varies linearly, with a variation across the bar is more than 10%, which is well above the acceptable one). This may be due to a slippage in one of the clamps. Again, in Fig. 15 (c), the strain fluctuation may be due to a local wrong alignment of the fiber.

In both cases,  it means that the authors do not fully control the setup.

An additional minor issue is the following: subplots should be always labeled following a clockwise sorting.
Answer 3: Agree, this was not pointed out precisely. For a tensile test in which both sides of the homogeneous specimen are hinged, no additional strains are to be expected. In this case, however, we have some uncertainties. Here the specimen is rigidly clamped, i.e. the supports can bear all internal forces (N, V, M, Mt). In addition, the samples have residual stresses from rolling and the temperature impact when cutting the grooves. Furthermore, varying cross-sectional dimensions are possible due to different cutting depths.

Referring to your 2 examples:

Fig. 12 cat. 3 left: a pre-curvature of the bar may be decisive here. By clamping the bar and pulling it, the bar experiences a bending moment in the centre, like a bending line. It is comparable to the imperfection ‘pre-curvature’.

Fig. 14 a/b (former: 15a/b): A time-dependent change such as a slippage of a clamp cannot be read from the linear local drift. At most, it could be interpreted within the wide data range of the data on a gauge. The linear drop can be explained by a displacement of the bar ends in relation to each other, the corresponding imperfection is ‘support displacement’

Fig. 14 c/d (former: 15c/d): The local misalignment of the fibre could be the decisive factor here. A combination of pre-curvature and varying cutting depth is also possible.

This is not ideal, unfortunately we only had this possibility to load the samples, and we did not have the possibility to control everything. We have already dealt with this in detail in Chapter 6, we have clarified the relevant paragraphs: L848ff, L865ff

We have also drawn up the following conclusion for the sensor rods: “Previous findings have shown that the uncertainties in manufacturing the sensor rods are so high that they are not well suited for precise strain measurement in manual production. Furthermore, a hinged installation of the specimens in the testing machine would be better.” (L877ff)

Furthermore, we rearranged the partial figures clockwise (Fig. 10, 14).

Round 2

Reviewer 3 Report

Comments and Suggestions for Authors

The authors amended the paper adequately according to the reviewer's suggestions. One minor suggestion: I suggest removing cat.1 from Figure 12 (it makes it awkward to visualize no-data), and I would rename cat. 1 into cat. 0 and all subsequent categories accordingly.

Author Response

Comments: One minor suggestion: I suggest removing cat.1 from Figure 12 (it makes it awkward to visualize no-data), and I would rename cat. 1 into cat. 0 and all subsequent categories accordingly. 

Response: Thank you very much for this idea. Due to the big issue of sensor defects, which is also an important criteria for the sensor's quality we would like to not change the figure and the categories. In our opinion, the description for category 1 (and subcategory 0) describes the issue of failing sensors very well. Furthermore, the admission of failures should not be considered awkward. The no-data visualisation is the logical illustration that completes figure 12.